# A long-term epigenetic memory switch controls bacterial virulence bimodality

Irine Ronin[1], Naama Katsowich[2], Ilan Rosenshine[2]*, Nathalie Q Balaban[1]*

[1]Racah Institute of Physics, Edmond J. Safra Campus, Faculty of Science, The Hebrew University of Jerusalem, Jerusalem, Israel; [2]Department of Microbiology and Molecular Genetics, IMRIC, Faculty of Medicine, The Hebrew University of Jerusalem, Jerusalem, Israel

**Abstract** When pathogens enter the host, sensing of environmental cues activates the expression of virulence genes. Opposite transition of pathogens from activating to non-activating conditions is poorly understood. Interestingly, variability in the expression of virulence genes upon infection enhances colonization. In order to systematically detect the role of phenotypic variability in enteropathogenic *E. coli* (EPEC), an important human pathogen, both in virulence activating and non-activating conditions, we employed the ScanLag methodology. The analysis revealed a bimodal growth rate. Mathematical modeling combined with experimental analysis showed that this bimodality is mediated by a hysteretic memory-switch that results in the stable co-existence of non-virulent and hyper-virulent subpopulations, even after many generations of growth in non-activating conditions. We identified the *per* operon as the key component of the hysteretic switch. This unique hysteretic memory switch may result in persistent infection and enhanced host-to-host spreading.

*For correspondence: ilanr@ ekmd.huji.ac.il (IRos); nathalie. balaban@mail.huji.ac.il (NQB)

Competing interests: The authors declare that no competing interests exist.

## Introduction

Bacterial populations spontaneously differentiate into distinct phenotypic states (*Avery, 2006*; *Dubnau and Losick, 2006*). This variability has been described as a bet-hedging strategy that results in subpopulations that will survive under unpredictable stress (*Fraser and Kaern, 2009*). It has also been suggested that phenotypic variability is a 'division of labor' strategy: essentially, the bacterial population diversifies in order to utilize nutrients more efficiently or to allow invasion and colonization of diverse niches in the host (*Ackermann et al., 2008*). Diversification upon infection is also related to antigenic variation, which is a key strategy to eluding the acquired immune response of the host (*Kamada et al., 2015*; *Stewart et al., 2011*).

The role of phase-variation mechanisms in phenotypic diversification through reversible genetic changes is well established (see for example, *Casadesús and Low, 2013*; *McClain et al., 1991*; *Silverman and Simon, 1980*). Diversification processes, not linked to DNA alteration have been attributed to noise in gene expression that can be further amplified by regulatory motifs such as excitatory dynamics (*Süel et al., 2006*) positive feedback leading to bi-stability (*Ozbudak et al., 2004*) and ultrasensitivity (*Temme et al., 2008*; *Rotem et al., 2010*; *Levine et al., 2012*). Interestingly, phenotypic diversification in microorganisms is frequently accompanied by growth rate variability. One striking example is that of bacterial persistence under antibiotic treatment (*Lewis, 2000*) mediated by growth rate bimodality (*Balaban et al., 2004*; *Brauner et al., 2016*; *Helaine and Holden, 2013*).

Pathogenic bacteria tightly regulate the expression of virulence machinery. Environmental conditions that are close to those in the host environment can induce the expression of the virulence genes ('activating conditions') (*Leverton and Kaper, 2005*; *Rosenshine et al., 1996*). In contrast,

**eLife digest** Bacteria typically cope with harsh and changing environments by activating specific genes or accumulating those mutations that change genes in a beneficial way. Recently, it was also shown that the levels of gene activity can vary between otherwise identical bacteria in a single population. This provides an alternative strategy to deal with stressful conditions because it generates sub-groups of bacteria that potentially already adapted to different environments. Bacteria that enter the human body face many challenges, and this kind of pre-adaptation could help them to invade humans and overcome the immune system. However, this hypothesis had not previously been tested in a bacterium called enteropathogenic *E.coli*, which infects the intestines and is responsible for the deaths of many infants worldwide.

Ronin et al. show that cells in enteropathogenic *E.coli* colonies spontaneously form into two groups when exposed to conditions that mimic the environment inside the human body. Once triggered, one of these groups is particularly dangerous and this "hypervirulent" state is remembered for an extremely long time meaning that the bacteria remain hypervirulent for many generations. In addition, Ronin et al. identified the specific genes that control the switch to the hypervirulent state.

These findings have uncovered the existence of groups of enteropathogenic *E.coli* that are pre-adapted to invading human hosts. Finding out more about how the switching mechanism works and its relevance in other bacteria may help researchers to develop new therapies that can help fight bacterial infections.

'non-activating conditions' include a broad spectrum of conditions that do not resemble the host environment. Variability in the expression of virulence genes is observed when the bacteria are exposed to activating conditions (*Nielsen et al., 2010*; *De Angelis et al., 2011*; *Somvanshi et al., 2012*; *Atack et al., 2015*). This process has been extensively studied in *Salmonella* (*Ackermann et al., 2008*; *Temme et al., 2008*). *Salmonella* employ a type III secretion system (T3SS) to inject the host cells with virulence factors. Interestingly, upon shifting from non-activating to activating conditions, *Salmonella* exhibit bimodal T3SS expression. The burden of T3SS expression, together with its bimodal expression, results also in growth rate bimodality (*Ackermann et al., 2008*; *Diard et al., 2013*; *Hautefort et al., 2003*; *Sturm et al., 2011*). The bimodality in the T3SS expression provides *Salmonella* with a fitness advantage in the host (*Diard et al., 2013*), contributing also to antibiotic persistence (*Arnoldini et al., 2014*), and to reduction in generation of non-virulent mutants termed 'defectors' (*Diard et al., 2013*).

The aim of this study was to examine whether phenotypic variability plays a role in the virulence of a model organism, enteropathogenic *E. coli* (EPEC), a human specific pathogen, during infection and in the transition to non-activating conditions. The major virulence factors of EPEC are a T3SS, similar to that of *Salmonella*, and a type IV pili termed the bundle forming pili (BFP) (*Gaytán et al., 2016*; *Hazen et al., 2016*, *2015b*). EPEC can cause symptoms ranging from asymptomatic infection to a devastating lethal disease in infants and spreads in the host population by the fecal-oral route (*Hazen et al., 2016*). Whereas the phenotypic variability of virulence upon shifting from non-activating to activating conditions has been extensively studied (e.g. *Arnoldini et al., 2014*; *Diard et al., 2013*; *Sturm et al., 2011*), the opposite process (i.e., the behavior of the pathogen population upon shifting from activating to non-activating conditions) is poorly understood. In vivo, shifts from activating to non-activating conditions can occur transiently within the host and also in the process of host-to-host spread.

Using our recently established ScanLag (*Levin-Reisman et al., 2010*) setup that can detect sub-populations lag time or growth rate variability by tracking single-colony appearance, we evaluated the phenotypic variability of growth of EPEC populations upon activation and in the transition from activating to non-activating conditions. Our analysis revealed a novel long-term hysteretic memory-switch in EPEC, which mediates bimodality in virulence expression. Whereas bimodal virulence expression is observed as a transient behavior in activating conditions, the transition from activating to non-activating conditions resulted in the stable co-existence of non-virulent bacteria and a

hypervirulent subpopulation that continued to express full virulence even after many generation of growth in non-activating conditions. It is likely that this hysteretic switch is common in pathogenic bacteria, ensuring persistence of infection and improved host-to-host spreading.

## Results

### Colonies originating from an activated EPEC culture exhibit bimodal growth

The expression of the EPEC virulence machinery is activated upon growth in Dulbecco's Modified Eagle's medium (DMEM) at 37°C to $OD_{600}$ of 0.2–0.5 ('activating conditions'). In contrast, overnight growth in Luria-Bertani liquid medium (LB) is considered as 'non-activating conditions' (*Hazen et al., 2015a*; *Leverton and Kaper, 2005*; *Puente et al., 1996*; *Rosenshine et al., 1996*). We searched for growth heterogeneity in EPEC upon transition from activating to non-activating conditions. As a negative control, the non-pathogenic *E. coli* K-12 was also evaluated. Cultures were grown under activating or non-activating conditions and plated on LB agar plates (i.e., non-activating conditions). Analysis of the colony growth dynamics, using ScanLag (*Levin-Reisman et al., 2010*) scanners, showed that EPEC and K-12 from the overnight LB cultures displayed unimodal distributions of colony appearance times (*Figure 1A,B*). In contrast, colonies of EPEC that originated from activating culture conditions exhibited a bimodal distribution of appearance time (*Figure 1C*, *Figure 1— source data 1*); the 'activated' K-12 culture maintained a unimodal distribution (*Figure 1D*). Further analysis showed that the bimodality in EPEC colony appearance time was due to a slightly reduced growth rate of the bacteria in the late-appearing colonies (*Figure 1—figure supplement 1*). These differences in the growth rates resulted in bimodal colony size distribution at 17 hr post-plating (*Figure 1E–G*). We refer to these two colony morphotypes as BIG and SMALL, for early- and late-appearing colonies, respectively (*Figure 1C,F*).

### SMALL morphotype is triggered by a resettable phenotypic switch

To examine the inheritability of the SMALL and BIG phenotypes, we resuspended SMALL and BIG colonies and immediately re-plated the bacteria on LB agar (*Figure 2A*). Most SMALL colony bacteria gave rise to SMALL colonies (96% ± 3%; mean ± s.d.) (*Figure 2B,D*), whereas the bacteria originated from BIG colonies gave rise to bimodal distribution in colony size with 79% ± 4% BIG and 21% ± 4% SMALL colonies (*Figure 2C,E*). Remarkably, repeating this procedure for four consecutive cycles resulted in similar ratios of BIG to SMALL colonies (*Figure 2D,E*). These findings suggested that the memory of the SMALL phenotype is inherited for tens of generations.

To test whether the switching between BIG and SMALL morphotypes is mediated by DNA rearrangement, we extracted DNA from BIG and SMALL colonies and sequenced the genomes at high coverage (*Supplementary file 1*). The genome sequences of SMALL and BIG colonies were identical and very close to that of the published reference sequence of EPEC strain E2348/69 (*Iguchi et al., 2009*). More advanced analysis, using a custom algorithm for the detection of DNA rearrangements typical to phase variation, identified three loci that were undergoing active phase variation by DNA inversion (*Goldberg et al., 2014*). However, we did not detect any specific differences in these regions between the BIG and SMALL genomes. The only significant difference between the two genomes was an approximately two-fold lower coverage of pMAR2 (EAF plasmid, *Iguchi et al., 2009*) in the BIG variant genome relative to the SMALL genome (*Supplementary file 1*). These results suggest that genetic changes are not involved in the colony size bimodality, favoring the possibility that the morphotypes are produced through an epigenetic mechanism.

We noticed a reduction in the SMALL morphotype inheritability when SMALL colonies were grown for more than 24 hr, before suspending and re-plating, suggesting that growth to stationary phase may affect the SMALL morphotype memory. To test this prediction, SMALL or BIG colonies were resuspended and grown in LB broth to stationary phase and then plated and analyzed. In both cases the SMALL morphotype disappeared (*Figure 2F–H*), and the culture was 'reset' to form the unimodal colony-size distribution typical of that reported for stationary-phase cultures (*Figure 1A*, *Levin-Reisman et al., 2010*). Taken together, these results show that the SMALL morphotype is extremely stable during growth, but disappears in stationary phase cultures.

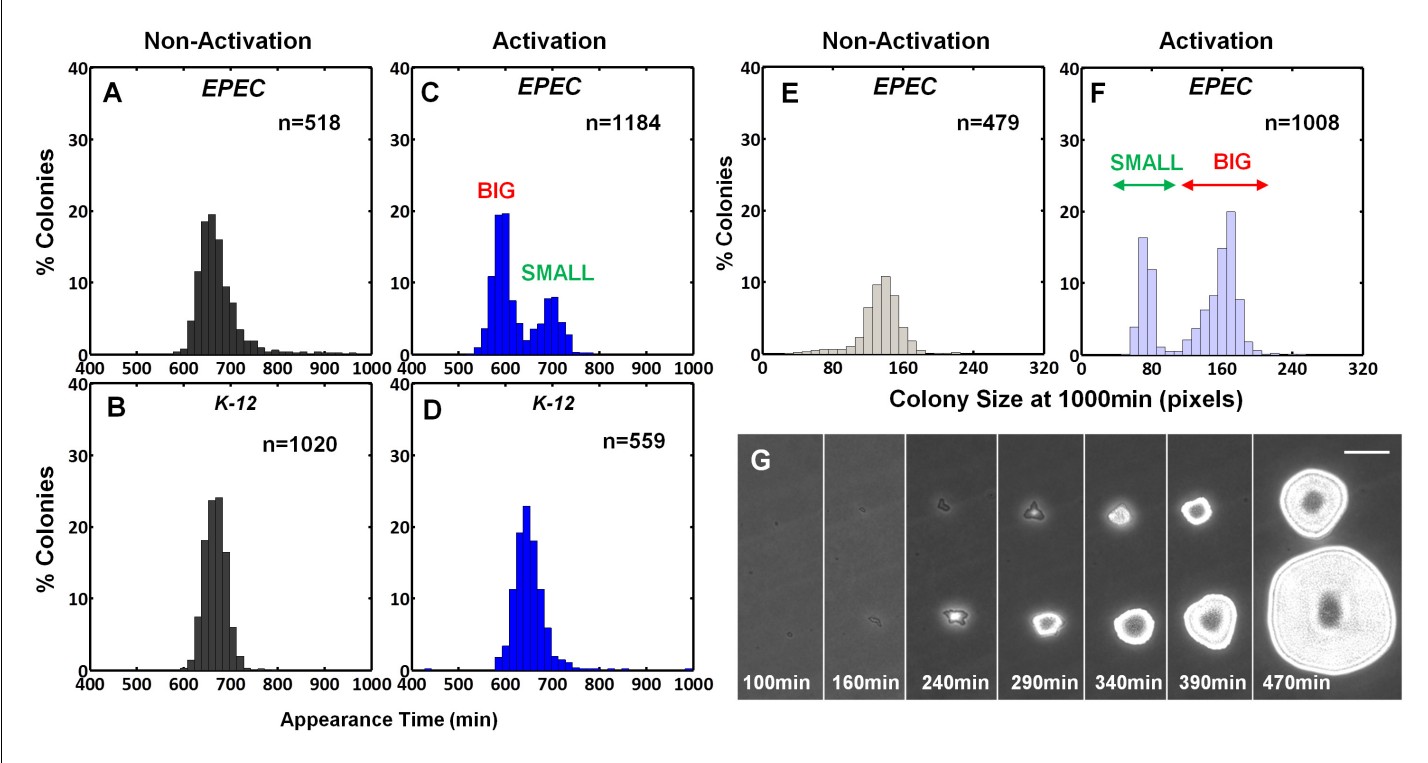

**Figure 1.** EPEC displays the bimodal colony size after virulence activation. Bacterial cultures of EPEC or *E. coli* K-12 were grown overnight in LB media (Non-Activation) or in DMEM for 3 hr at 37°C to OD$_{600}$ ~0.3 (Activation). Cultures were then diluted, plated on LB agar, and incubated at 32°C (non-activating conditions). Colony appearance time was monitored by ScanLag at 15 min intervals. The resulting histograms show (A, B, C, D) the fractions of colonies detected at each time point and (E, F) colony size distributions 1000 min after plating. Colonies larger or smaller than 105 pixels were defined as 'BIG' and 'SMALL' morphotypes, respectively. (A–C, E, F) Experiments were repeated in at least four independent biological replicates. (D) shows the cumulative of 4 independent biological replicates. (G) Time-lapse microscopy phase-contrast images of the two co-existing morphotypes, BIG and SMALL. Time points are indicated. Similar results were obtained in at least 10 different locations and in two independent biological replicates. Scale bar: 50 μm.

The following source data and figure supplement are available for figure 1:

**Source data 1.** This Source Data file contains appearance time histogram raw data for *Figure 1A and C* (activated and non-activated EPEC cultures) from ScanLag experiments.

**Figure supplement 1.** EPEC growth on LB measured by ScanLag and time-lapse microscopy.

## Mathematical analysis of the bimodal switch in non-activating conditions

In order to characterize the switching process and measure the switching rates between the two morphotypes, we fitted the results of the SMALL and BIG colony fractions in non-activating conditions to a model based on switching between two phenotypes with different growth rates (*Balaban et al., 2004*) (*Figure 3A*, *Equations 1 and 2*, see Materials and methods-Mathematical model). BIG bacteria (*B*) have a higher growth rate and can switch to the SMALL morphotype with rate *a*, whereas SMALL bacteria (*S*) can switch to the BIG morphotype with rate *b*. The model reproduced our experimental observations and suggested that upon growth in non-activating conditions, the switching rate from BIG to SMALL is about 10 times higher than from SMALL to BIG, resulting in a stable co-existence of the two morphotypes. Furthermore, the time scale of the switching from SMALL to BIG was extremely long, requiring more than 100 hr and many generations to reach steady state (see Materials and methods-Mathematical model).

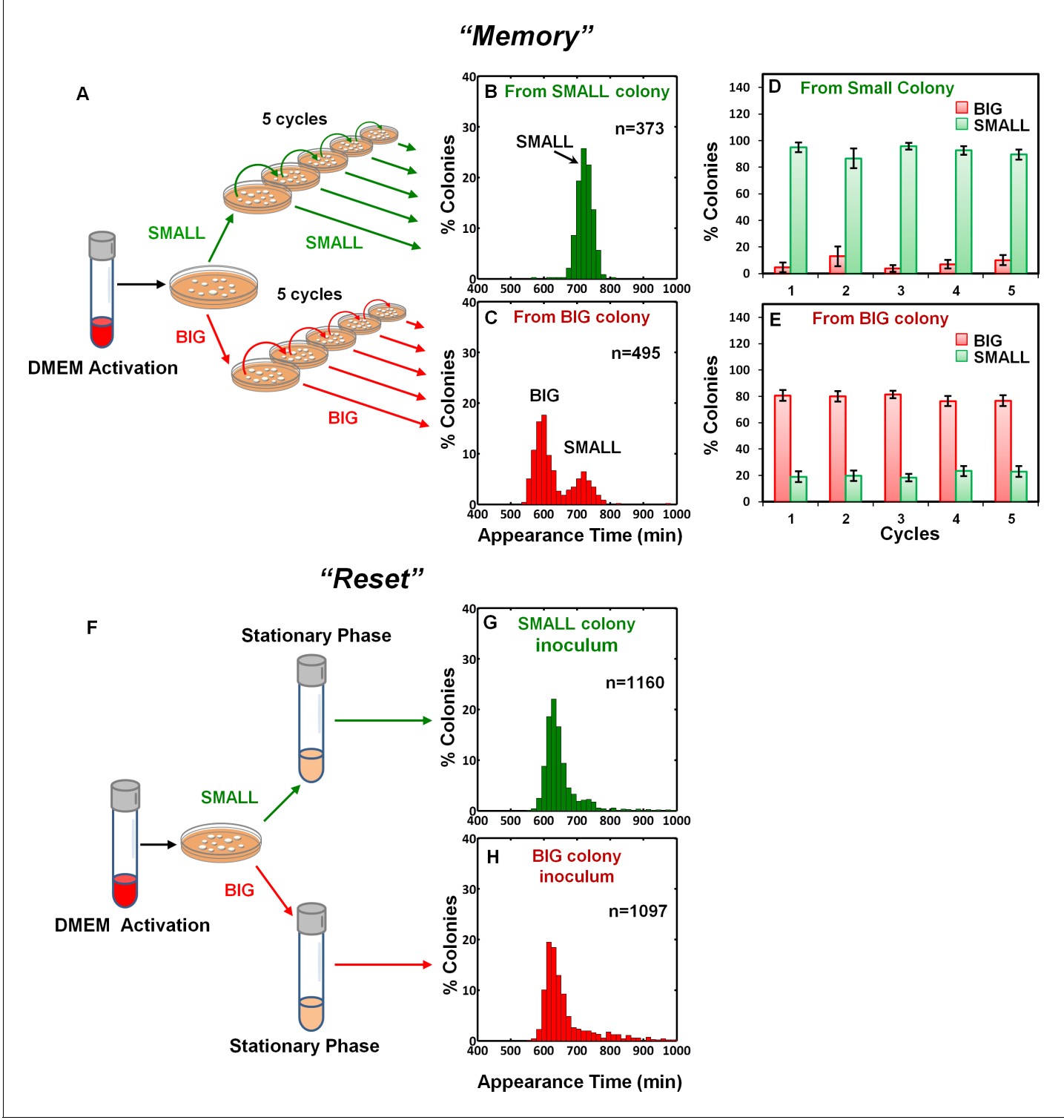

**Figure 2.** Memory and reset of colony-size bimodality. (**A**) Scheme of the experimental procedure for measuring the stability of colony morphotypes: Colonies of EPEC grown on LB agar were picked 1000 min after plating, resuspended, re-plated on LB agar, and subjected to ScanLag analysis. (**B**) Histogram of the fraction of colonies detected at each time point for bacteria taken from a SMALL colony. (**C**) Histogram of the fraction of colonies detected at each time point for bacteria taken from a BIG colony. (**B–C**) Experiments were repeated in at least four independent biological replicates. (**D, E**) The same procedure was repeated for four consecutive cycles using bacteria taken from (**D**) SMALL or (**E**) BIG colonies, and in each cycle the fraction of BIG and SMALL colonies was determined. Data are presented as the means ± s.d. of five technical replicates. (**F**) Scheme of the experimental

*Figure 2 continued on next page*

*Figure 2 continued*

procedure for the 'reset' of the bimodality. (G) SMALL or (H) BIG colonies of EPEC were picked 1000 min after plating, resuspended in LB broth and grown overnight to stationary phase at 37°C. Cultures were then plated and subjected to ScanLag analysis.

## Prediction and measurements of switching dynamics in activating conditions

The switching rates extracted from the growth of EPEC bacteria in non-activating conditions result in slow dynamics of switching between the two morphotypes. However, already after 3 hr of growth in activating conditions, we observe a high proportion of SMALL morphotypes, suggesting that the switching rate from BIG to SMALL in activating conditions is higher than in non-activating conditions. Our model predicted that longer exposure to activating conditions would lead to a higher percentage of the SMALL morphotype. In order to evaluate the predictive value of the model and the switching rates under activating conditions, we diluted a stationary phase culture into DMEM and monitored BIG and SMALL colony ratios over time. We found that the switching from BIG to SMALL was tenfold faster under activating conditions than under non-activating conditions (*Figure 3B* and Materials and methods-Mathematical model). As a result, within a few hours under activating conditions, the SMALL morphotype dominated the culture. Similarly to growth in LB, stationary phase resets the culture to unimodality (*Figure 3B,C*). Taken together, these results show that although the bimodal switch generates variability both under activating and non-activating conditions, the higher switching rate under activating conditions results in a fast increase of the SMALL to BIG ratio. In both conditions, stationary phase resets the culture to a unimodal BIG population.

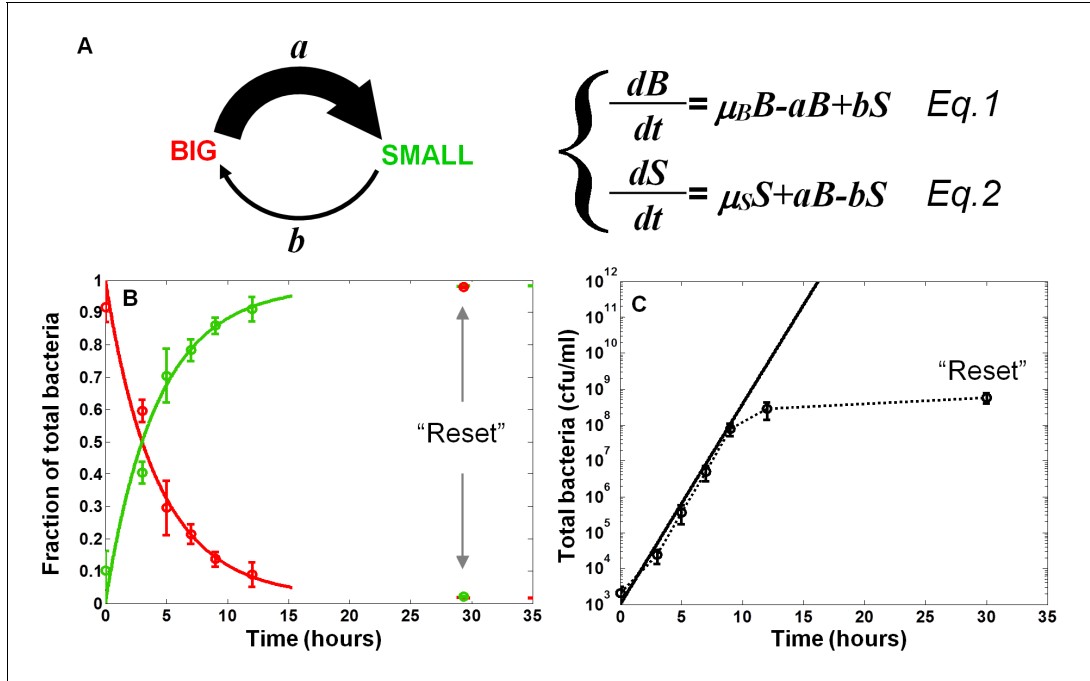

**Figure 3.** Model and measurements of bimodal switching rates. (A) Scheme and equations of a bimodal switching model. The two morphotypes, BIG and SMALL, are characterized by different growth rates, $\mu_B$ and $\mu_S$, respectively, and different switching rates $a$ and $b$. Note that these parameters depend on growth conditions. (B) Measurement and fit to the analytical solution of equations *Equations 1 and 2* during exponential growth under activating conditions with initial conditions $B(t = 0)=1$, $S(t = 0)=0$, see Materials and methods-Mathematical model. Green and red lines are ScanLag measurements of the SMALL and BIG morphotype fractions, respectively (means ± s.d. of three independent biological replicates). Solid lines are the fit to data using *Equations 1 and 2*, resulting in $a = 0.24 \pm 0.13$ h$^{-1}$ and $b<<a$ under activating conditions. These switching rates result in a population dominated by the SMALL morphotype after a few hours. Note that stationary phase caused resetting of the culture to the BIG morphotype. (C) Model (solid line) and experimental measurement (dotted line and markers) of the growth of the total population for the data presented in (B).

## PerA and PerB are sufficient for the bimodal phenotype

Activating conditions, which were shown above to increase the SMALL/BIG ratio, are known to activate the expression of key transcriptional regulators of EPEC virulence including Ler, GrlA, PerA, and PerC (*Figure 4A*, reviewed in *Clarke et al., 2003*; *Mellies et al., 2007*). We tested for possible involvement of these regulators in the BIG to SMALL switch and found that Ler and GrlA are not required for the bimodal colony size phenotype (*Figure 4—figure supplement 1*). In contrast, the EPEC strain cured of the EAF plasmid, which encodes the *perABC* operon, lost the capacity to generate bimodality and produced only BIG colonies (*Figure 4B*). Complementing this strain with the EAF plasmid restored the bimodal phenotype. To identify the EAF plasmid genes required to establish the bimodality, we examine mutant strains *ΔperA*, *ΔperC*, and *ΔbfpA* (*Figure 4B*, *Figure 4—figure supplement 1*). Notably, only the *ΔperA* mutant failed to exhibit bimodality. Importantly, PerA is the autoactivator and thus the *perA* mutant is deficient in expressing the entire *perABC* operon. Complementing the *ΔperA* strain with a low copy number plasmid, containing the *perABC* operon with its native regulatory region transcriptionally fused to *gfp* (pPerABC-GFP)restored the bimodality. In this case, the SMALL colonies appeared later and were more abundant than in the *wt* strain, probably due to excessive PerABC expression. These results show that the *per* operon is required for the co-existence of the BIG and SMALL morphotypes, whereas PerC, GrlA, Ler, T3SS biogenesis, and BFP formation were not required for colony size bimodality (*Figure 4—figure supplement 1*). To find whether PerA or PerB underlie the bimodality, we deleted different fragments from the pPerABC-GFP plasmid resulting in pPerA-GFP, pPerB-GFP or pPerAB-GFP. Notably, we kept the native regulatory region implying that in all cases PerA is required for expression. We transformed the *ΔperA* mutant with plasmids expressing PerA-GFP, PerB-GFP or PerAB-GFP. Whereas complementation of *ΔperA* with PerA-GFP or PerB-GFP expressing plasmids resulted in unimodal colony morphotypes (*Figure 5A*), the PerAB-GFP plasmid restored bimodality, indicating that both PerA and PerB are required for the bimodality. Microscopic observations show that GFP expression is bimodal in pPerAB-GFP plasmids, while uniformly high in pPerA-GFP (*Figure 5B*). PerB and GFP cannot be expressed from pPerB-GFP in the absence of PerA and accordingly no GFP was observed in *ΔperA* mutant containing this plasmid (*Figure 5*). Expression of GFP from this plasmid was restored in the *wt* EPEC (*Figure 5—figure supplement 1*). These results show that both PerA and PerB expression are required for the co-existence of the BIG and SMALL morphotypes. To determine whether other EAF plasmid factors are required for the growth bimodality, we transformed *E. Coli* K-12 MG1655 strain with the above pPer plasmids and found that co-expression of PerA and PerB is sufficient for induction of bimodality also in this strain. In agreement with the results obtained in the *ΔperA* EPEC strain, PerA or PerB alone failed to generate bimodality (*Figure 5*). These results show that PerAB expression generates a bimodality of growth also without the EAF plasmid, although the phenotype was milder (i.e. the growth difference between the two morphotypes is smaller) (*Figure 5—figure supplement 2*).

## Bimodality of *perABC* expression during activation underlies colony-size bimodality

We next asked whether the colony-size bimodality correlates with bimodality in *perABC* expression in the progenitor cells, namely the founders of each colony. We subjected a culture of the *ΔperA* EPEC complemented with pPerABC-GFP to activating conditions and measured GFP expression by microscopy (*Figure 6A*). We detected a bimodal expression of the *perABC-gfp* operon. Furthermore, the bacteria that did not express GFP were larger and divided more rapidly than those that expressed GFP. To correlate between *perABC-gfp* expression and colony size, we used Fluorescence-activated cell sorting (FACS) to collect separately the GFP-OFF and GFP-ON bacteria (*Figure 6B*). Each subpopulation was then plated and analyzed by ScanLag (*Figure 6C,D*). The results showed that the GFP-OFF bacteria grew in BIG colonies, whereas the GFP-ON bacteria generated almost exclusively SMALL colonies. Similar results were obtained using wild-type EPEC transformed with pPerABC-GFP, whereas no bimodality was observed with control pZS11*GFP plasmid expressing GFP from a synthetic constitutive promoter (*Figure 6—figure supplement 1*). Taken together, these results show that during growth in activating conditions, a bimodal expression of the *perABC* operon is established in progenitor cells that lead to bimodality in colony size upon plating and growth in non-activating conditions.

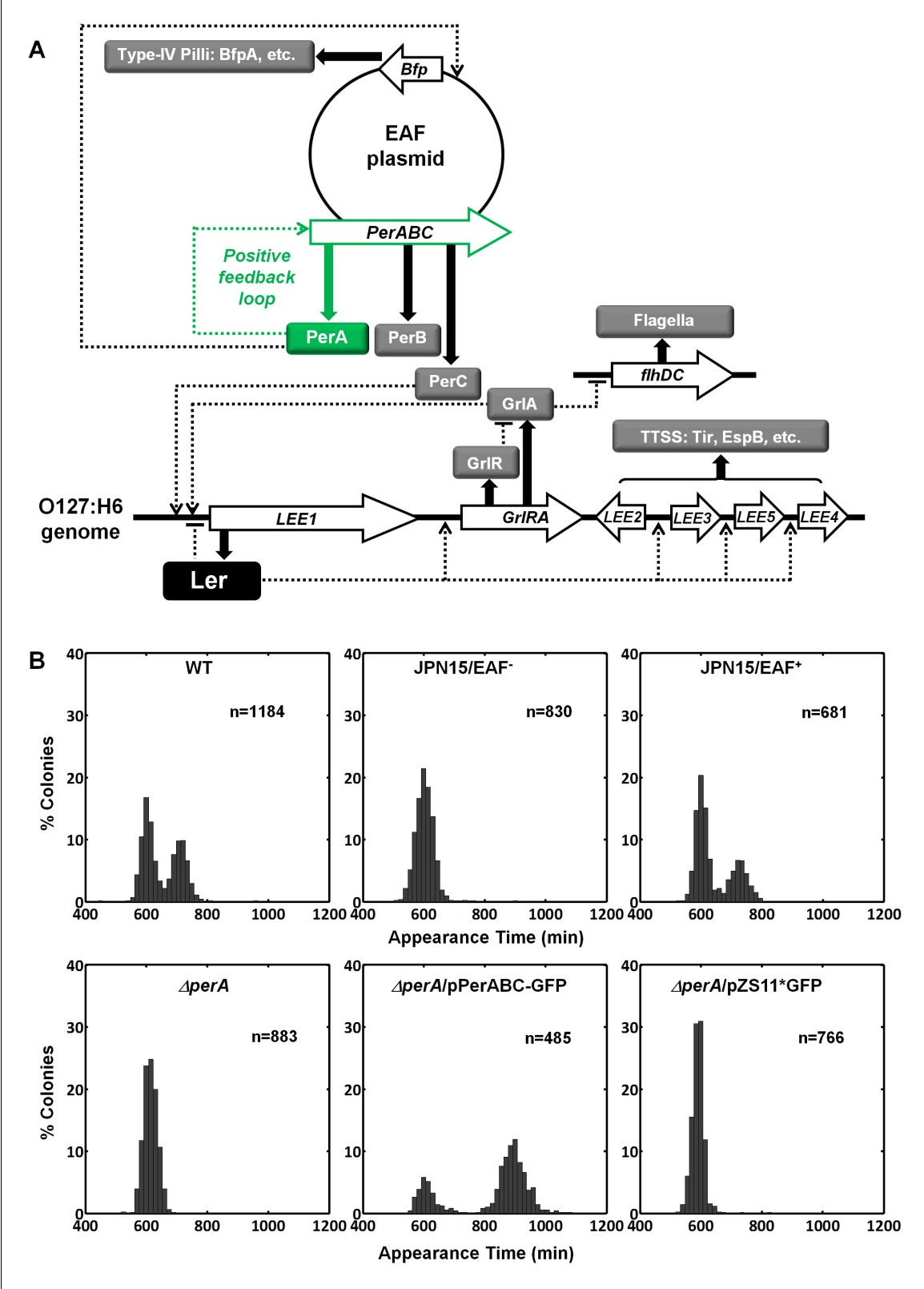

**Figure 4.** Per operon is essential for establishing colony-size bimodality. (**A**) Scheme of key regulatory genes of the EPEC virulence machinery. Ler is the T3SS master regulator and its expression is induced by two redundant positive regulators, PerC and GrlA (*Bustamante et al., 2011*). PerA is a positive autoregulator (*Ibarra et al., 2003*; *Martínez-Laguna et al., 1999*; *Porter et al., 2004*) of *perABC* operon and positive transcription regulator of typeIV pilli (bfpA) (*Ibarra et al., 2003*; *Tobe et al., 1996*). Open arrows represent operons, thick arrows and filled boxes represent protein production. Dotted

*Figure 4 continued on next page*

*Figure 4 continued*

lines indicate regulatory circuits. PerA positive feedback loop is marked in green. (**B**) Histogram of colony appearance times for bacteria taken from activated cultures of indicated strains. Strains without *perA*, either by gene deletion (*ΔperA*) or EAF plasmid curing (JPN15/EAF⁻), result in unimodality. These experiments were repeated in at least two independent biological replicates.

The following figure supplement is available for figure 4:

**Figure supplement 1.** ScanLag colony appearance phenotype of EPEC virulence pathway mutants.

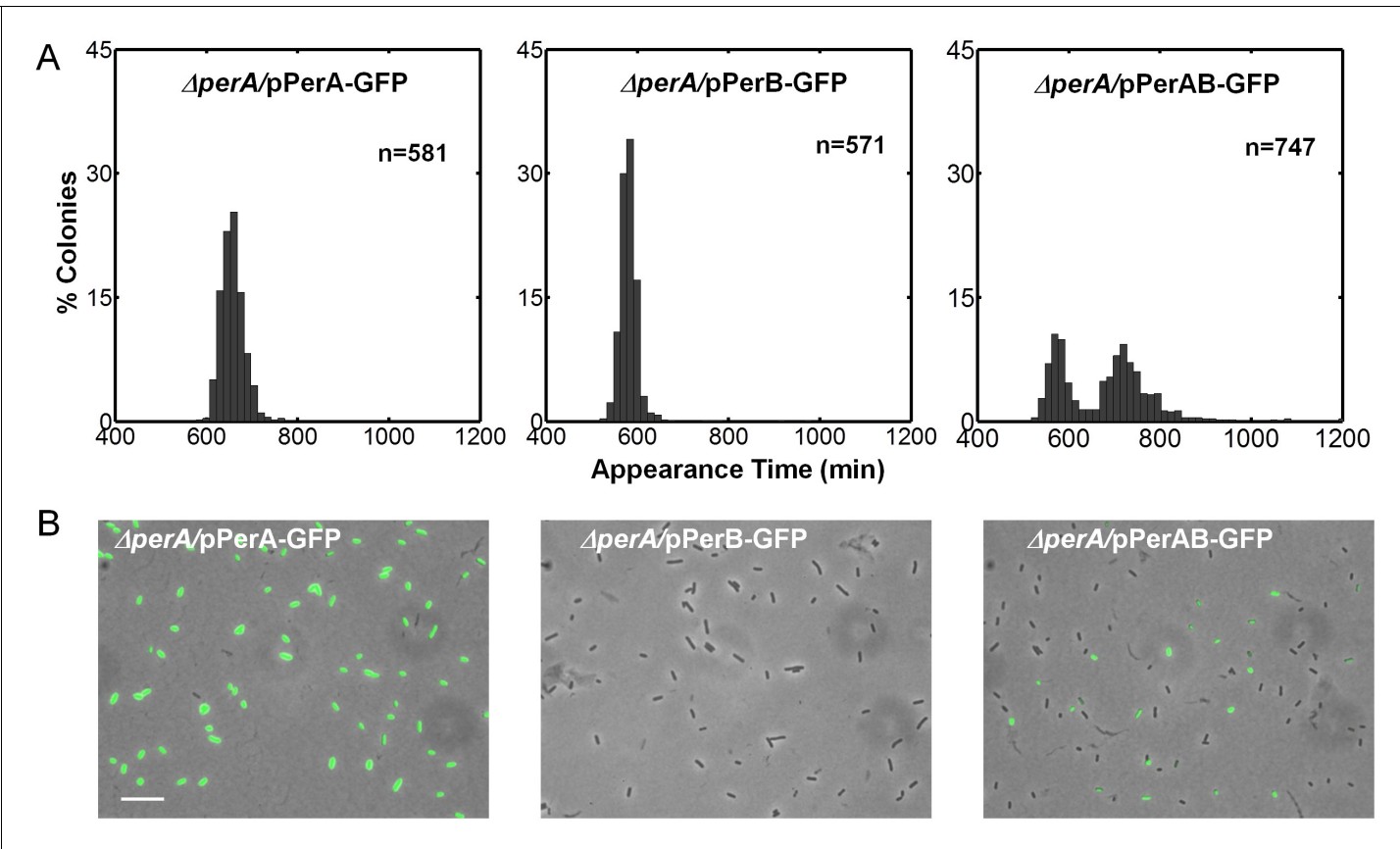

**Figure 5.** Co-expression of PerA and PerB results in colony-size bimodality. (**A**) A histogram of the fraction of colonies detected at each time point for bacteria taken from indicated cultures of EPEC *ΔperA* mutant transformed with pPerA-GFP (*ΔperA*/pPerA-GFP), pPerB-GFP (*ΔperA*/pPerB-GFP) and pPerAB-GFP plasmids (*ΔperA*/pPerAB-GFP). Cultures were started from a single colony (BIG) and grown in activating conditions. Only pAB-GFP plasmid, co-expressing PerA and PerB, restored the bimodality. This experiment was repeated in at least two independent biological replicates. (**B**) Expression of *perA*, *perB* and *perAB* using a transcriptional GFP reporter by time-lapse microscopy of single cells extracted from BIG colonies as in (**A**) collected 1000 min after plating. Similar results were obtained in two independent biological replicates and in at least three different locations. Scale bar: 15 μm.

The following figure supplements are available for figure 5:

**Figure supplement 1.** Expression control of pPerB-GFP in *wild type* EPEC.

**Figure supplement 2.** Co-expression of PerA and PerB causes bimodality of colony growth in *E. Coli* K-12 MG1655.

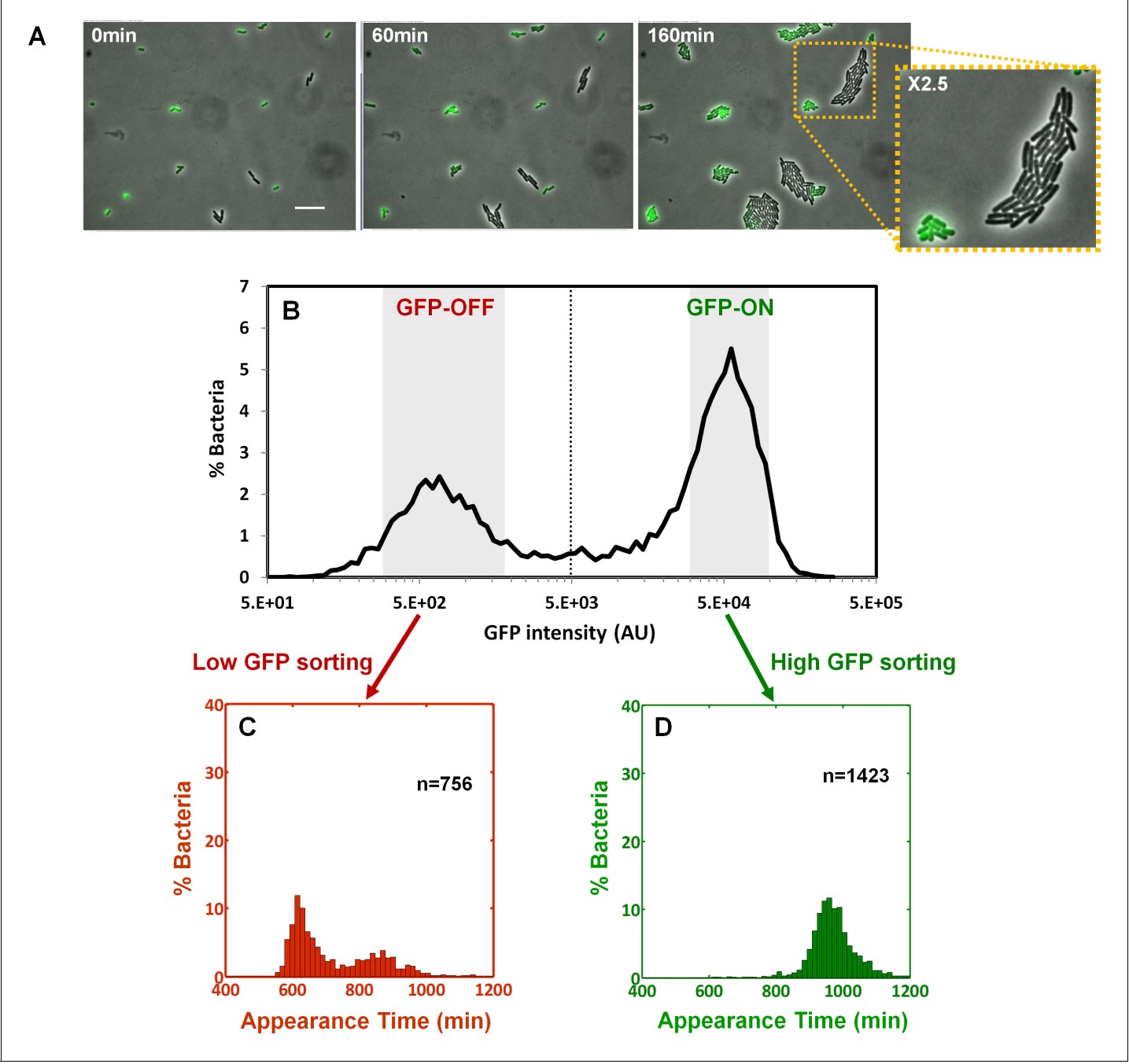

**Figure 6.** Bimodality of *perABC* expression during activation underlies colony-size bimodality. EPEC *ΔperA* containing the plasmid pPerABC-GFP was grown under activating conditions. (**A**) Time-lapse microscopy of the activated *ΔperA*/ pPerABC-GFP under non-activating conditions (i.e. on LB-agar pads, at 32°C) Scale bar: 15 μm. (**B**) Flow cytometry analysis (*t* = 0 min) for levels of GFP in the cells (n = 10000 bacteria). Time points are indicated. Similar results were obtained in at least five different locations and in two independent biological replicates. (**C–D**) Sorted fractions of *perABC* GFP-ON and GFP-OFF (from **B**) populations were plated under non-activating conditions and analyzed by ScanLag. Histograms show appearance time of sorted (**C**) GFP-OFF and (**D**) GFP-ON subpopulations. (**B–D**) Experiments were repeated in at least two independent biological replicates.

The following figure supplement is available for figure 6:

**Figure supplement 1.** Expression of *perABC* in wild-type EPEC during activation.

## Bimodality of *perABC* expression results in bimodal *ler* expression upon shift from activating to non-activating conditions

Ler is the T3SS master regulator, and under activating conditions its expression is induced by two redundant positive regulators, PerC and GrlA (*Bustamante et al., 2011*; *Gómez-Duarte and Kaper, 1995*; *Porter et al., 2004*) (*Figure 4A*). Under activating conditions, we observed that Ler expression from chromosomal *ler-gfp* transcriptional fusion was unimodal (*Figure 7A–C*), as previously reported (*Berdichevsky et al., 2005*; *Roe et al., 2004*). This was in contrast to the bimodal *perABC-gfp* expression in the same conditions (*Figure 6A,B*). We predicted, however, that upon shifting the culture from activating to non-activating conditions, which suppresses GrlA activity, only the sub-population that expresses *perABC* will continue to express *ler,* resulting in bimodal Ler expression. As predicted, we found that upon shifting from activating to non-activating conditions, Ler-GFP expression was reduced in approximately 50% of the bacterial population and became bimodal (*Figure 7B,D*). Time-lapse microscopy of bacteria taken from BIG and SMALL colonies showed that, as expected, Ler expression was uniformly high in the SMALL population but bimodal in the BIG population (*Figure 7—figure supplement 1*). These results suggest that the bimodal expression from *perABC* during activating conditions does not result in bimodal Ler expression since GrlA, which is redundant to PerC, activates *ler* expression regardless of whether PerC is expressed or not. However, upon shifting to non-activating conditions and the bimodal expression of *perABC* drives the bimodality of Ler expression.

## Long-term memory of the virulence state is controlled by the *per* operon through a hysteretic switch

The stability of the SMALL morphotype suggests that the virulence switch mediated by the *per* operon is a hysteretic switch, maintaining long-term memory of the previous state as shown in *Figure 8A*. Accordingly, the entire population switches to a virulent state upon activation, characterized by unimodal and high *ler* expression (*Figure 8A*, State 1). In contrast, *per* expression during activation is typically bimodal, although prolonged activation eventually shifts the population to SMALL (*Figure 3B*). When transferred to non-activating conditions, the *per*-ON bacteria remain hypervirulent, expressing both Ler and PerABC, leading to SMALL colony morphotype (*Figure 8*, State 2). This hypervirulent state is maintained for an extremely long time but is 'reset' to the non-virulent state during stationary phase (*Figure 8*, State 3). When a 'reset' population is subjected to non-activating conditions the majority remains in the BIG morphotype (*Figure 8*, State 4). This insight leads to several predictions. Firstly, the SMALL morphotype should express virulence factors downstream of both *ler* and *per* (i.e., BFP and T3SS), even after many generations of growth in non-activating conditions. Secondly, the BIG colony morphotype should consist mainly of non-virulent bacteria with ~20% of hypervirulent ones. Finally, the deletion of the *per* operon should not prevent the activation of virulence (i.e., T3SS expression can be driven through the GrlA-Ler path), but erase the hysteretic switch, and thus *ler* expression and virulence of all bacteria, should decline as soon as the Δ*perA* bacteria are transferred to non-activating conditions (*Figure 8B*).

In order to test these predictions, we extracted proteins from EPEC and Δ*perA* cultures grown under conditions that would lead to states shown schematically in *Figure 8A,B* and performed Western blot analysis using antibodies raised against BfpA, EspB, and Tir. BfpA, an important constituent of the type IV pili, was used as readout for PerA activity, and the T3SS proteins EspB and Tir were used as readout for Ler activity. As an additional negative readout of virulence, we used flagellin (FliC$_{H6}$), which is known to be repressed by GrlA (*Iyoda et al., 2006*) (*Figure 4A*). As expected, in EPEC, EspB, Tir, and BfpA, but not FliC were highly expressed upon growth in activating conditions (*Figure 8A,C*: State 1). Similar expression patterns were seen in SMALL colony bacteria, despite the fact that at least 20 generations had passed since the transition from activating to non-activating conditions (*Figure 8A,C*: State 2). In contrast, bacteria from BIG colonies showed low expression levels of all virulence factors and high levels of FliC. The residual expression of BfpA, EspB, and Tir by the BIG colony bacteria is consistent with the prediction of ~20% SMALL variants in the BIG population (*Figure 8A,C*: State 4). Finally, the Δ*perA* mutant showed high expression of EspB and Tir when grown in activating conditions, consistent with activation of *ler* expression by GrlA (*Figure 8B,D*: State 1). However, expression of these proteins rapidly diminished when the mutant culture was shifted to growth under non-activating conditions (*Figure 8B,D*: State 4), leading to a pattern

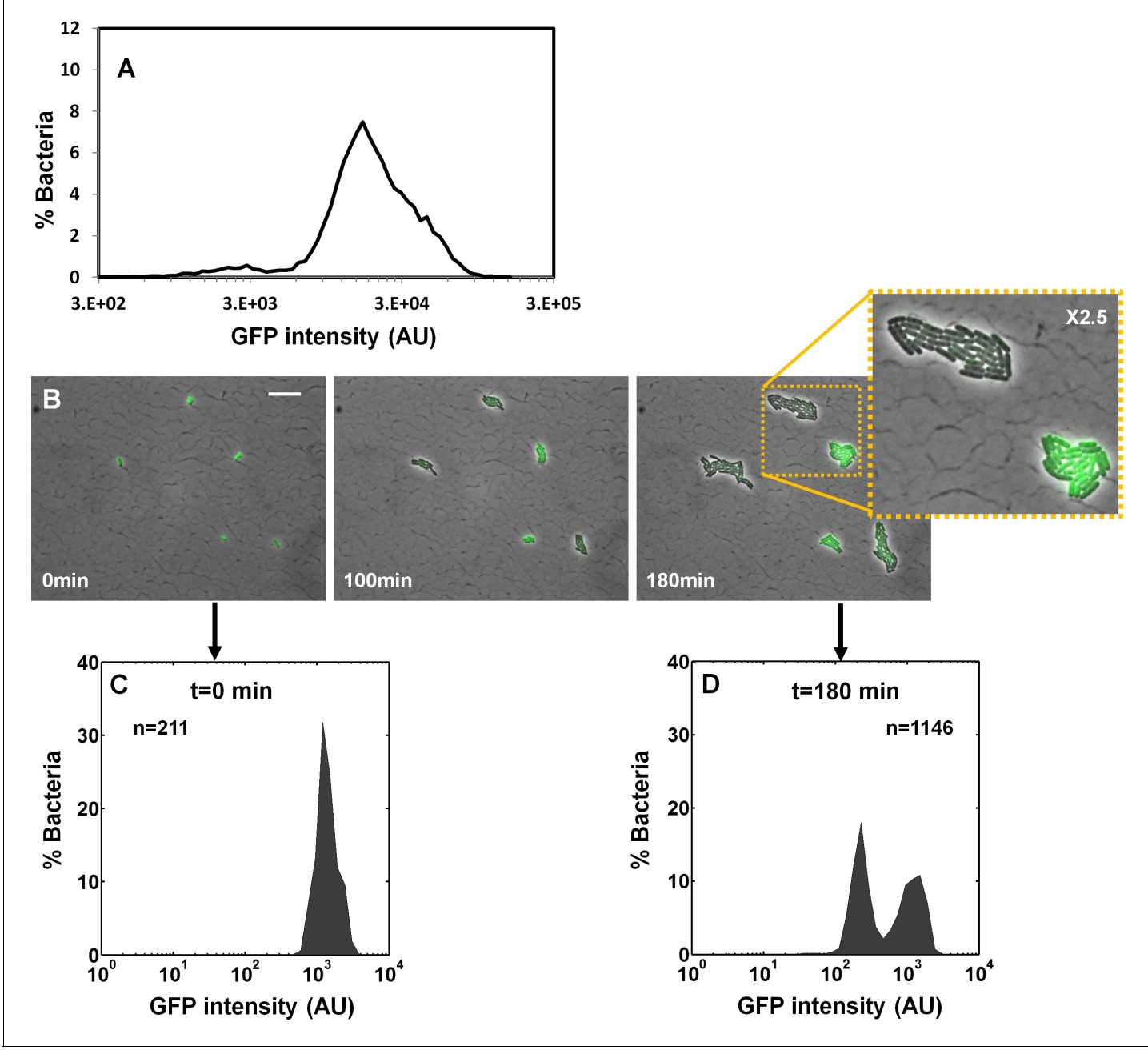

**Figure 7.** Ler expression is unimodal during activation but becomes bimodal when cells are shifted to non-activating conditions. EPEC *ler-gfp* was grown under activating conditions. (**A**) Flow cytometry analysis (*t* = 0 min) shows unimodal GFP (Ler-ON state) expression (*n* = 10000). (**B**) Time-lapse microscopy during growth under non-activating at indicated times. Scale bar: 15 μm. (**C–D**) Quantification of GFP levels from the images shown in panel (**B**). Similar results were obtained in at least five different locations and in two independent biological replicates.

The following figure supplement is available for figure 7:

**Figure supplement 1.** *Ler* is differentially expressed in BIG and SMALL colony morphotypes.

expected from a unimodal non-virulent population (i.e., high FliC, low EspB and low Tir) and similar to the pattern of expression in the stationary phase LB culture (*Figure 8A,C*: State 3).

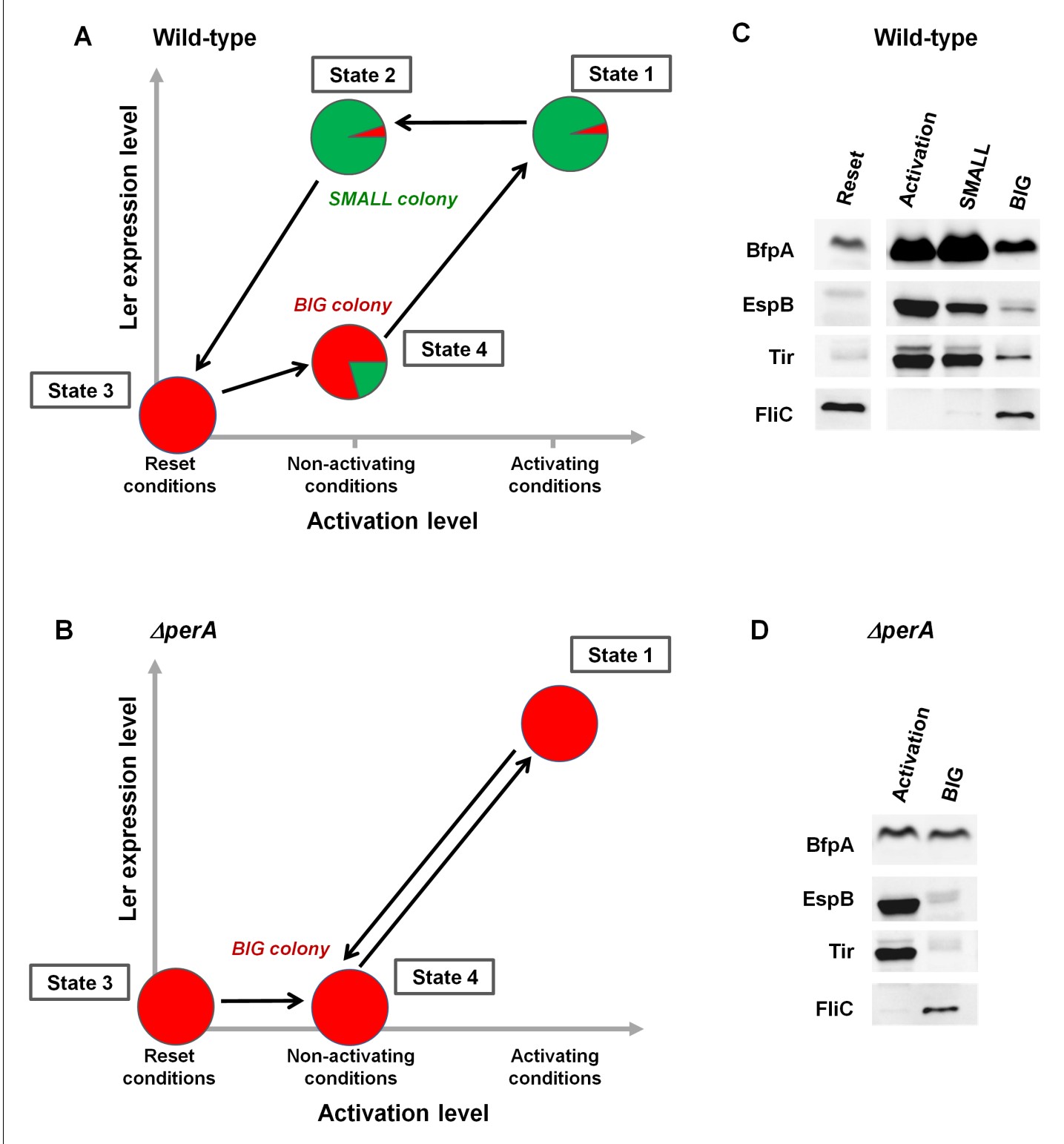

**Figure 8.** PerABC maintains long-term memory through a hysteretic switch. (**A**) Scheme of the hysteretic switch in wild-type EPEC. Subjecting a culture to activating conditions for several hours results in a majority of SMALL bacteria (green) (State 1) (*Figure 3B*). Even when transferred to non-activating conditions, the SMALL bacteria maintain their phenotype (State 2), unless subjected to stationary phase conditions results in BIG (red) colonies (State3, 'Reset'). Growth under non-activating conditions maintains a majority of the BIG phenotype (State 4). Shifting again to activating conditions regenerates the SMALL phenotype (State 1). (**B**) Deletion of *perA* abolishes the hysteretic switch but does not prevent *ler* activation. (**C**) Western blot analysis of proteins extracted from wild-type EPEC in the different states defined in (**A**) using antibodies raised against BfpA, EspB, Tir, and FliC$_{H6}$. The following

*Figure 8 continued on next page*

Figure 8 continued

conditions were used: Reset: LB overnight culture (State3); Activation (State1); SMALL colony (State2); BIG colony (State 4). (D) Western blot analysis of proteins extracted from EPEC *ΔperA* grown in Activation (State 1) and BIG colony (State 4). (C,D) Similar results were obtained in at least two independent biological replicates.

The following figure supplement is available for figure 8:

**Figure supplement 1.** Loading control of proteins for Western blot analysis.

## Bimodality in *per* operon expression results in bimodality of host cells infectivity

Our results show that the SMALL colonies express a high level of functional T3SS and BFP and thus may be hypervirulent. To determine whether this expression pattern results in a hypervirulent phenotype, we first tested SMALL colony bacteria for BFP functionality by monitoring BFP-mediated self-aggregation (*Bieber et al., 1998*). Time-lapse microscopy showed that, as expected, resuspended SMALL colony bacteria rapidly aggregated, whereas BIG colony bacteria remained mostly planktonic (*Figure 9A*, *Video 1* (BIG), *Video 2* (SMALL), *Figure 8—figure supplement 1A*). In both cases, the aggregates disintegrated upon reaching stationary phase, consistent with the resetting of colony-size bimodality (*Figure 2F–H*) and the significant reduction in BfpA production in stationary phase (*Figure 8C*).

In order to determine whether high levels of *per* operon expression correlate with higher infectivity, we followed the infection of Hela cells with EPEC *ΔperA* bacteria transformed with pPerABC-GFP plasmid. We used microscopy to evaluate *perABC-gfp* expression, aggregation (i.e., microcolony formation), and attachment to host cells. The results showed that the bacteria that expressed PerABC (GFP-ON, *per*-ON) formed microcolonies that rapidly attached to the epithelial cells (*Figure 9B*). In contrast, the GFP-OFF (*per*-OFF) bacteria remained mostly planktonic and unattached. These results show that the *per*-ON bacteria, which generate the SMALL colony morphotype, display higher infectivity than *per*-OFF bacteria.

We next asked whether this high infectivity is maintained in bacteria taken from SMALL colonies (i.e., bacteria originating from *per*-ON bacteria but that were grown in non-activating conditions for many generations). We infected HeLa cells with a 1:1 mix of bacteria from SMALL and BIG colonies tagged with constitutive YFP and mCherry (*Gefen et al., 2008*), respectively, and compared infectivity by time-lapse microscopy (*Figure 9C*, *Figure 9—figure supplement 1B*). During the first hours of infection, the SMALL bacteria formed aggregates (microcolonies) and almost no planktonic single bacteria were found in the surrounding medium (*Figure 9C–E*, *Figure 9—figure supplement 1B*, *Videos 3* and *4*). Furthermore, these SMALL microcolonies attached to host cells, an indication of BFP function (*Figure 9—figure supplement 2*), induced actin rearrangement in the host, and invaded into the host cell, indicatives of T3SS functionality (*Figure 9—figure supplement 2*, *Figure 9—figure supplement 3*). In contrast, only a few bacteria that originated from BIG colonies were organized into attached microcolonies, and formation of actin rearrangement as well as invasiveness were marginal (*Figure 9C–E*, *Figure 9—figure supplement 1B*, *Videos 3* and *4*, *Figure 9—figure supplement 3*). These results show that the SMALL morphotype maintains high infectivity and expresses functional BFP and T3SS even after many generations of growth in non-activating conditions.

## Discussion

This study showed, for the first time to our knowledge, that the virulence machinery of a human pathogen, EPEC, is controlled by a hysteretic switch with long epigenetic memory. We showed that PerA and PerB are sufficient for this hysteretic switch. We found that when exposed to virulence-activating conditions all EPEC bacteria upregulate expression of T3SS virulence genes, unlike the bimodal virulence expression observed in *Salmonella*. However, we found that the EPEC virulent population is bimodal for expression of the *per* regulated genes, resulting in two coexisting virulent sub-populations of bacteria, planktonic (*per*-OFF) and aggregative (*per*-ON), with different infection

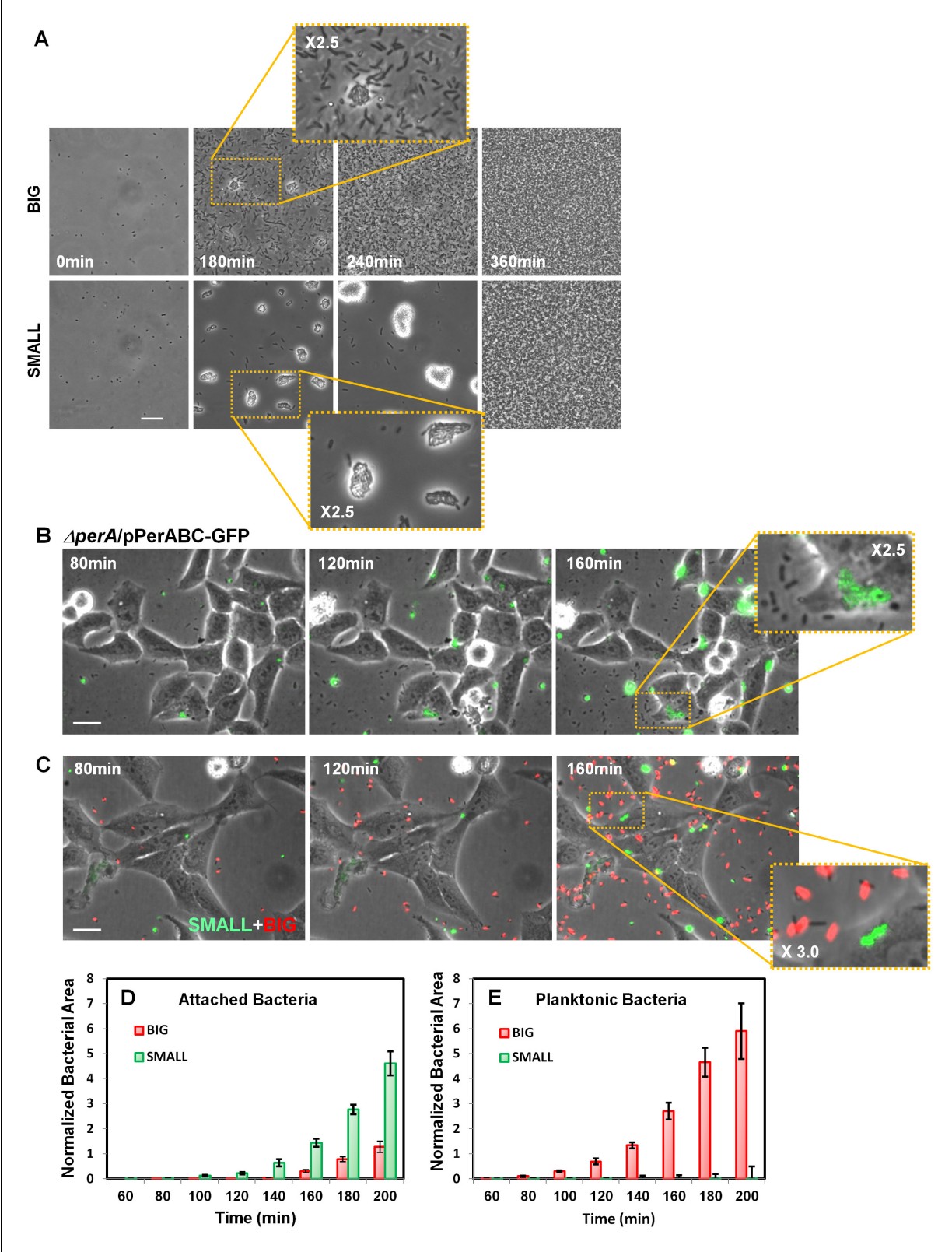

**Figure 9.** Bimodal *perABC* expression correlates with bimodality in microcolony formation and host cell attachment. (**A**) Phase-contrast images of the dynamics of self-aggregation observed by time-lapse microscopy of bacteria from BIG or SMALL colonies. Scale bar: 15 µm. See also *Videos 1* and *2*. (**B**) Time-lapse microscopy of HeLa cells infected with EPEC *ΔperA*/pPerABC-GFP. Scale bar: 25 µm. (**C**) Fluorescent time-lapse microscopy of HeLa cells infected with a 1:1 mixture of wild-type EPEC from BIG and SMALL colonies tagged with mCherry and YFP, respectively. Scale bar: 25 µm.

*Figure 9 continued on next page*

*Figure 9 continued*

Enlarged image shows SMALL bacteria attached to the Hela cells whereas the BIG bacteria are planktonic resulting in a shift between phase-contrast and red fluorescent image. (D, E) Quantification of (D) attached and (E) planktonic bacteria in images taken from (C). The area of bacteria was determined based on fluorescent signal. The attached bacteria area was normalized to the total area of HeLa cells in the frame. Planktonic bacteria area was normalized to the area free of cells. Data are presented as the means ± s.d. of 6 frames. The experiment was repeated three times. See also *Videos 3* and *4*. Similar results were obtained in at least five different locations and in two independent biological replicates.

The following figure supplements are available for figure 9:

**Figure supplement 1.** Bacteria from SMALL colonies have enhanced self-aggregation properties in liquid culture and increased formation of microcolonies on host cells.

**Figure supplement 2.** Bacteria from SMALL colonies induce massive pedestal formation during HeLa infection.

**Figure supplement 3.** Invasion of HeLa cells by EPEC from SMALL and BIG colonies.

and invasion abilities. The latter population constitutively expresses both BFP and T3SS for many generations, rapidly attaches to host cells as microcolonies, delivers effectors into the host cell, and invades it. This rapid invasion may protect the bacteria against the host immune system and establish persistent infection (*Tuchscherr et al., 2011*).

The striking difference between the two phenotypes that we unveiled is in their abilities to maintain their virulence state when transferred to non-activating conditions. Whereas the *per*-OFF bacteria no longer express *ler* regulated genes upon transfer to non-activating conditions, the *per*-ON bacteria maintain high expression of the *ler* regulated virulence genes even after tens of generations of growth in non-activating conditions. This *per*-ON long-term memory may allow the pathogen to overcome transit through niches of non-activating conditions without a drop in virulence level. An extreme case of this type of transit is the host-to-host transit through the fecal-oral route. In addition, the long-term memory *per*-ON state may set the stage for further diversification within this subpopulation, possibly creating a range of infective phenotypes, each adapted to a different niche and/or stage of infection within the host intestine. Alternatively, the hysteretic switch that enables the coexistence of two different phenotypes may be attributable to bet-hedging (i.e. to risk spreading in the absence of a predictable environment). The SMALL morphotype infects host cells more rapidly but bears the cost of expressing virulence genes and exposure to the immune system, whereas the BIG morphotype grows faster and is less immunogenic. Notably, bimodality is stable mainly under non-activating conditions, suggesting that insufficient cues from the environment regarding bacterial residence inside or outside the host may promote a bet-hedging strategy (*Kussell and Leibler, 2005*).

We found that PerA is central for this switch and that the PerA-regulated *perABC* operon exhibits bimodal expression when co-expressed with PerB. Notably, the *per* operon is regulated by PerA auto-activation (*Figure 4A*), a network motif that has been shown to lead to bimodality (*Smits et al., 2006*) and hysteresis (*Mitrophanov and Groisman, 2008*), which can lead to bistability by growth feedback mechanisms (*Deris et al., 2013*; *Irwin et al., 2010*;

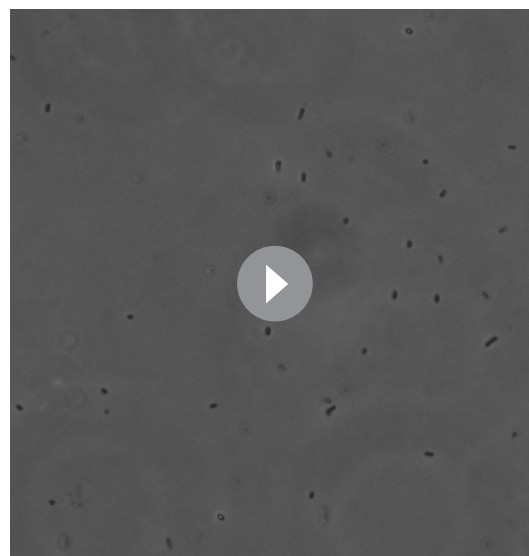

**Video 1.** Dynamics of self-aggregation observed by time-lapse microscopy in BIG bacteria. Bacteria were resuspended from a BIG colony and placed on a wet LB agarose pad for imaging bacteria in suspension. Bacteria divide and remain mostly planktonic.

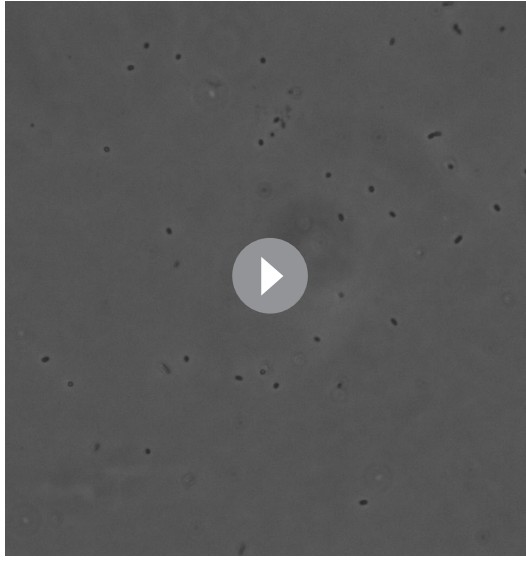

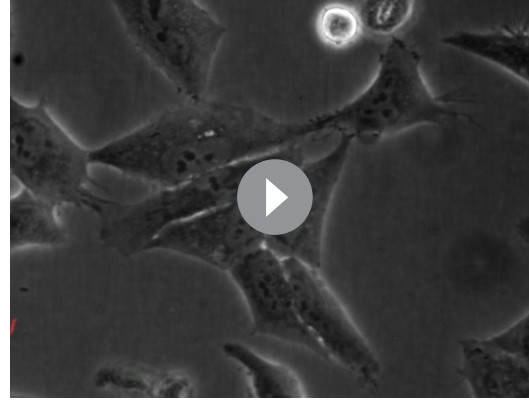

**Video 3.** Dynamics of infection by SMALL (green) and BIG (red) bacteria on HeLa cells. The SMALL bacteria form microcolonies attached to the HeLa cells, whereas the BIG bacteria remain mostly planktonic (same as *Figure 9C*).

**Video 2.** The dynamics of self-aggregation observed by time-lapse microscopy in SMALL bacteria. Bacteria were resuspended from a SMALL colony and placed on a wet LB agarose pad for imaging bacteria in suspension. Bacteria divide and aggregate continuously until they reach stationary phase, which results in the disintegration of the aggregates.

*Tan et al., 2009*), or other mechanisms, for which PerB may be required.

Specific environmental conditions (termed here 'activating conditions') strongly increased the *per*-ON frequency in the population by enhancing the switching rate by a factor of ~10 compared to non-activating conditions. Thus, growth under activating conditions resulted in almost 100% of *per*-ON cells at steady state. Importantly, our results show that *per* expression is essential for establishing a hysteretic long-term memory switch, resulting in the co-existence of *per*-OFF and *per*-ON subpopulations, of which the latter remains stable even in the face of drastic changes in the environmental conditions, such as shifts in temperature and growth media. Given this stability, a single bacterium in the *per*-ON state generates a colony of the SMALL morphotype in which most of the bacteria remain *per*-ON and are primed for rapid infection of host cells. Interestingly, the *per*-ON memory vanished once growth reaches stationary phase, and the entire population switched to *per*-OFF.

Bimodality in expression of virulence genes has been extensively studied in *Salmonella* and thus it is useful to compare the two pathogens. In EPEC, expression of the Ler master regulator, and thus expression of T3SS, is high and unimodal under activating conditions. Redundant activation of the *ler* promoter by independent regulators (GrlA and PerC) ensures that all bacteria express virulent genes during activation. In contrast, *Salmonella* displays bimodal T3SS expression upon growth under activating conditions and during initial infection (*Hautefort et al., 2003*; *Sturm et al., 2011*). The *per*-switch acts hysteretically, locking the expression of *ler* and activation of virulence in an 'ON' state, even when bacteria are switched back to non-activating conditions. Thus in EPEC,

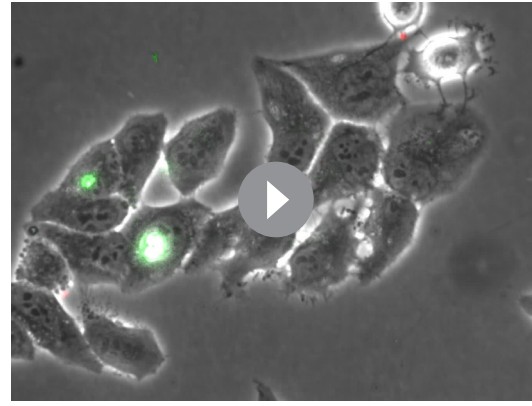

**Video 4.** Dynamics of infection by SMALL (red) and BIG (green) bacteria on HeLa cells. The SMALL bacteria form microcolonies attached to the HeLa cells, whereas the BIG bacteria remain mostly planktonic (same as *Video 3* but with fluorescent markers opposite tagging).

*ler* expression becomes bimodal only when the bacteria are transferred to non-activating conditions, resulting in the co-existence of non-activated bacteria and bacteria that are already primed for infection through constitutive expression of BFP and T3SS. Importantly, we were able to evaluate the rates of switching between the two EPEC phenotypes and found that the rate from ON to OFF is extremely slow (several weeks), even under non-activating conditions. A long-lived ON state has been observed also in *Salmonella* (*Sturm et al., 2011*), but upon shifting to non-activating conditions, regulators of virulence decay within 1 to 2 hr (*Temme et al., 2008*).

The importance of the described hysteretic switch for EPEC virulence is reflected by the conservation of the switch core: the *perABC* operon. Recent reports comparing the genome sequences of a large number of EPEC clinical isolates show that EPEC is an umbrella name for a collection of *E. coli* strains belonging to diverse phylogenetic branches that acquired independently, through horizontal gene transfer, a pathogenicity island encoding T3SS (the LEE island) (*Ingle et al., 2016*). Notably, most of these EPEC strains also acquired plasmids containing the *perABC* operon (*Ingle et al., 2016*). Furthermore, strains containing both the LEE and plasmids encoding *perABC* and *bfp* operons cause a more severe disease (*Hazen et al., 2015b*). Hazen et al. proposed that the contribution of PerA might be related to regulation of additional virulence-related genes (*Hazen et al., 2015b*). Our findings suggest that *perABC* also enhances the fitness of infecting EPEC by facilitating formation of long-term memory and stable phenotypic bimodality. The contribution of the *per*-switch to virulence in vivo could not be tested since an animal model for EPEC is not available. Pathogens closely related to EPEC, including enterohemorragic *E. coli* (EHEC) and *Citrobacter rodentium* (CR), do not carry the *perABC* operon. Interestingly, however, heterogeneity in virulence is observed in these pathogens (*Roe et al., 2004*; *Kamada et al., 2015*), but the involved switch and whether it is also hysteretic have not been studied.

In conclusion, we report here how a hysteretic switch controls the virulence traits of a human pathogen, EPEC. Our findings and approach should provide a framework to search for similar switches in other pathogens. Furthermore, this understanding may lead to the development of new strategies to interfere with the establishment of stable virulence-ON mechanisms and thus reduce virulence and pathogen spreading to new hosts.

## Materials and methods

### Bacterial strains and growth conditions

The used bacterial strains, plasmids and primers are listed (*Supplementary file 2* and *3*).

Activating conditions: Following the procedure in (*Kenny et al., 1997*), bacterial strains were grown overnight (O/N) in LB medium (Sigma, Israel) at 37°C without shaking, diluted 1:40 into DMEM-HEPES (Gibco, Israel) medium and incubated for 3 hr at 37°C without shaking to exponential phase (O.D.~0.3).

Non-activating conditions: bacteria were plated in LB agar at a concentration below 200 cfu/plate and incubated at 32°C.

For analysis of bacteria isolate directly from colonies, BIG and SMALL colonies were collected according to their size at 17 hr (1000 min) after plating and diluted in 0.9% NaCl to density of ~$10^8$ bacteria/ml.

### Plasmid construction and gene inactivation

Deletion mutants were produced as described (*Datsenko and Wanner, 2000*). pZS*GFP was created by replacing the *hip* promoter of pZS*1HGFP plasmid (*Rotem et al., 2010*) with the synthetic $P_{LtetO-1}$ promoter (*Lutz and Bujard, 1997*) using 5'-phosphorylated PCR primer, followed by ligation. pPerABC-GFP was created with an isothermal cloning kit (NEB, United States). The *hip* promoter of pZS*1HGFP was replaced with the genomic region of the *perABC* operon including the upstream promoter region. The derivative plasmids: pPerA-GFP, pPerB-GFP and pPerAB-GFP were produced by single ligation step after excision of the PerAB, PerA/C and PerC respectively (see *Supplementary file 2*).

## HeLa cell growth conditions

HeLa cells (*Supplementary file 2*) were grown in DMEM supplemented with 300 µg/ml L-Glutamine, 100 U/ml Penicillin, 100 µg/ml Streptomycin and 10% FCS at 37°C and 5% $CO_2$. For infection experiments, HeLa cells were seeded in 24-well plates. When cultures reached ~$10^6$ cells/well, they were washed twice with PBS and medium was changed to DMEM-HEPES (Gibco, United States) without supplements. HeLa cells were routinely tested for absence of mycoplasma contamination by EZ-PCR Mycoplasma Test Kit (Biological Industries LTD., Israel).

## Colony growth distribution measurements by ScanLag

Bacteria were diluted to a concentration of $10^3$ bacteria/ml and plated on LB agar. The plates were placed in a 32°C incubator on EPSON Perfection 3490 scanners that scan the plates every 15 min with custom ScanLag software, as described (*Levin-Reisman et al., 2010*). MatLab based applications were used to automatically detect colonies in each frame and to monitor the growth of individual colonies (Software for controlling the scanners and for image analysis can be found at http://bio-site.phys.huji.ac.il/Materials). The area growth rate of each colony and its time of appearance were extracted as described (*Levin-Reisman et al., 2010*)

## Experimental measurements of bimodal switching rates

EPEC was grown overnight in LB medium (Sigma, Israel) at 37°C, diluted to ~1000 cells/ml in DMEM and grown under activating conditions. The low culture density enabled follow up of many hours of exponential growth before reaching stationary phase. To eliminate possible artifacts due to the BFP-mediated-aggregation we induced disaggregation as follows: the culture was divided into tightly closed Eppendorf tubes, one for each time point. These tubes were subjected to intensive vortex and kept on ice for 10 min before plating. Disaggregation was confirmed by microscopy. The levels of BIG and SMALL progenitors in the populations at each time point during growth were determined by ScanLag.

## Time-lapse microscopy

Time-lapse microscopy was performed using a Leica DMIRE2 inverted microscope system. Autofocus and image acquisition were done by using custom macros in µManager (an open source software program) to control the microscope, stage, shutters, and camera. The microscope was placed in a large incubator box (Life Imaging Systems) that controls the temperature to an accuracy of 0.1°C. GFP- or YFP-expressing bacteria were imaged using Yellow GFP filter (Ex-500nm, Em-535nm, Chroma USA); mCherry signal was measured by HcRed1 filter (Ex-575nm, Em-640nm, Chroma USA). Excitation was performed with LEDs (Coolled, United Kingdom) and images were acquired with a cooled CCD camera ($-75$°C) (Orca II, Back-illuminated, Hamamatsu) and processed with ImageJ (http://rsbweb.nih.gov/ij/). X100 NA 1.40 oil objective was used for individual bacterial observation on agar pads; X63 NA 0.70 long-distance air objective NA for imaging Hela and bacteria in 24-well plates; X20 was used for imaging growing colonies on a very thin LB + 1.5% agar layer.

To observe the growth of individual bacteria, a LB + 1.5% agarose pad was prepared in a polydi-methylsiloxane (PDMS) square mold and dried for 10 min at 37°C. Bacterial samples of 1 µl (~$10^5$ cells) were placed between the microscopic slide and the pad inserted into the same PDMS mould and covered with a coverslip. For microscopic observation of self-aggregation, agarose pads were not dried and bacteria (5 µl, ~$10^5$ cells) originated from suspended colony were placed between the microscopic slide and the pad.

## Western blot analysis

The OD of the cultures was determined and density was adjusted if needed. Bacteria were collected by centrifugation and resuspended in 5 µl loading sample buffer per 0.1 O.D. Proteins were extracted by boiling of the samples and resolved by 12% Mini-PROTEAN TGX Stain-Free Precast Gels (Bio-Rad, Israel). Total protein staining was used as a loading control (see *Figure 8—figure supplement 1*). Proteins were transferred to nitrocellulose membrane (Bio-Rad) for standard Western blot analysis with antibodies raised in rabbits against BfpA (gift from Michael Donnenberg), Tir and EspB (gift from Gad Frankel), or FliC-H6 (gift from the Israeli Ministry of Health) and secondary anti-rabbit HRP-conjugated antibody.

## EPEC self-aggregation test

Bacteria were diluted 1:50 in LB media in 24-well plates. The plates were incubated at 32°C with mild agitation under non-activating conditions and transferred manually every 30 min to an Epson Perfection V500 Scanner for imaging of aggregates.

## Actin staining for microscopy

HeLa cells were grown on round coverslips within 24-well plates. In the next day cells were washed and infected with ~$10^6$ bacteria/well. Two hours after infection, the wells were washed twice with PBS and fixed with 4% formaldehyde in PBS, 10 min. The coverslips were then washed twice with TBS, and the cells were permeabilized with 0.25% Triton X-100 for 2 min. Actin was stained with Texas Red-phalloidin (Molecular Probes, United States) and DNA was stained with Dapi (Molecular Probes, United States), at a 1:1000 dilution. The coverslips were washed twice with PBS, mounted with ImumMount (Thermo Scientific, United States) and viewed with a fluorescence inverted microscope with X100 oil objective.

## Time-lapse imaging of HeLa cell infection

HeLa cells were seeded in 24-wells plates (NUNC, United States) and grown in 1 ml/well as mentioned above. EPEC BIG/pZA21mCherry and SMALL/pZA21YFP colonies (harboring mCherry and YFP constitutive markers), were suspended, mixed 1:1 and added to the HeLa cells at a concentration of ~$10^6$ bacteria/well. Time-Lapse microscopy was performed directly on 24-wells plates (NUNC, United States) at 37°C. The opposite fluorescent markers (i.e. SMALL/pZA21mCherry and BIG/pZA21YFP) were measured in parallel to rule out effects of markers. The analysis of fluorescent intensities was performed with ImageJ. The total area occupied by bacteria was measured according to the yellow fluorescence signal. In order to compare different frames with variable Hela cell coverage, we normalized the total number of bacteria attached to cells with the total area of cells in the frame. This area was extracted from phase-contrast images. Planktonic bacteria area was normalized with the total area of the frame not covered by Hela cells.

## Gentamicin protection assay

HeLa cells grown in 24-well plates and infected with EPEC bacteria were incubated at 37°C for 2 hr. Gentamicin at a final concentration of 25 µg/ml was added to kill extracellular bacteria with little effect on intracellular bacteria (*Benjamin et al., 1995*) and plates were returned to the 37°C incubation. Bacteria that survived the gentamicin treatment were counted at 30 min intervals after gentamicin addition. To this end cells were washed twice to remove gentymicine, lysed in 1 ml 1% Triton X-100 to free the intracellular bacteria, which were then pelleted by centrifugation (3 min at 1500g), resuspended in PBS and spread onto LB agar plates to evaluate bacterial CFU/ml. Isolated colonies were counted after overnight incubation at 37°C, and the progeny of BIG-YFP and SMALL-mCherry colonies were differentiated by color. The time point t = 0 sample was measured before gentamicin treatment. All bacterial counts were normalized to counts at t = 0 counts. The same results were obtained with swapped reporter colors, SMALL-YFP and BIG-mCherry.

## Direct visualization of live internalized bacteria

EPEC containing a plasmid with IPTG inducible GFP reporter (pSA11) was used to generate SMALL colonies, which were suspended and used to infect HeLa cells. Two hours post infection gentamicin was added (25 µg/ml) and after an additional hour IPTG (20 µg/ml) was added to the infected cells. Only the metabolically active intracellular bacteria that are not exposed to gentamicin are able to produce GFP in response to IPTG, thus enabling to detect even a small fraction of intracellular bacteria by time-lapse microscopy.

## Immunostaining and confocal microscopy imaging

For immunostaining experiments, the gentamicin protection assay was performed as described above, except for the growth of HeLa which was here done on round coverslips inserted into 24-well plates. HeLa cells were infected with EPEC SMALL/pSA11 bacteria for 2 hr and then treated with 25 µg/ml gentamicin. After 1 hr wells were supplied with IPTG to visualize metabolically active intracellular bacteria. One hour later samples were washed twice with PBS and fixed with 4% formaldehyde-

PBS. The coverslips were washed twice with TBS, the cells were permeabilized with 0.25% Triton X-100 and washed twice with PBS. Actin was stained with Texas Red-phalloidin (Molecular Probes) and DNA was stained with Hoechst (Molecular Probes, United States), at a 1:1000 dilution. After washing twice with PBS, the coverslips were mounted with ImumMount (Termo Scientific, United States) and viewed with a FV-1200 Olympus (Japan) confocal microscope. The GFP signal was measured with a green filter (Ex-500nm, Em-540nm), mCherry signal was measured with a red filter (Ex-570nm, Em-620nm) and Hoechst was measured with a Dapi filter (Ex-430nm, Em–470nm).

## Fluorescence-activated cell sorting (FACS)

BIG or SMALL colonies were suspended, diluted in PBS to ~$10^5$ cells/ml and analyzed using a FACS Aria III cell-sorter equipped with 488 nm and 561 nm lasers (BD Biosciences, San Jose, CA). Side and forward scatter of bacteria were determined using log scale SSC/FSC plots with respective thresholds of 200 and 2200. Sorting was done at a minimal flow rate according to GFP intensity criteria.

## Whole genome sequencing (WGS)

BIG and SMALL colonies were suspended, diluted 1:200 and bacteria were grown in LB to O.D.~0.3 at 32°C. DNA was extracted with the DNeasy Blood and Tissue kit (Qiagene) according the manufacturer's instructions. Genomic extraction, Whole-Genome Sequencing and analysis was done as published previously (*Goldberg et al., 2014*). The WGS raw data are available as NCBI BioProject PRJNA255355 (Accessions: SRX757584 and SRX757585 for SMALL and BIG respectively)

## Mathematical model

We used a simple mathematical model to describe the expected dynamics of switching between the BIG and SMALL morphotypes. Fitting the experimental results to the data enabled the evaluation of the switching rates, *a* (BIG to SMALL) and *b* (SMALL to BIG), under virulence activating conditions. Surprisingly, we found that what was considered as 'non-activating conditions', namely growth in LB, does not abolish the switching but rather reduces its frequency.

The two morphotypes, BIG (*B*) and SMALL (*S*) are characterized by different growth rates, $\mu_B$ and $\mu_s$ respectively, and switching rates, *a* and *b*, respectively (*Equations 1 and 2* in *Figure 3A*). The analytical solution of *Equations 1 and 2*, as detailed in (*Balaban et al., 2004*), is:

$$B(t), S(t) = e^{\bar{\mu}t}\left[\alpha_{B,S}e^{\Omega t} + \beta_{B,S}e^{-\Omega t}\right]$$

$$
\begin{cases}
\mu_B^* &= \mu_B - a \\
\mu_S^* &= \mu_S - b \\
\bar{\mu} &= \frac{\mu_B^* + \mu_S^*}{2} \\
2\Omega &= \sqrt{\left(\mu_B^* - \mu_S^*\right)^2 + 4ab}
\end{cases}
\begin{cases}
\alpha_B &= \frac{B_0\left(\bar{\mu} + \Omega - \mu_S^*\right) + S_0 b}{2\Omega} \\
\beta_B &= \frac{-B_0\left(\bar{\mu} - \Omega - \mu_S^*\right) - S_0 b}{2\Omega}
\end{cases}
\begin{cases}
\alpha_S &= \frac{S_0\left(\bar{\mu} + \Omega - \mu_B^*\right) + B_0 a}{2\Omega} \\
\beta_S &= \frac{-S_0\left(\bar{\mu} - \Omega - \mu_B^*\right) - B_0 a}{2\Omega}
\end{cases}
\tag{1}
$$

where $B_0$ and $S_0$ are the numbers of the BIG and SMALL morphotypes at *t* = 0.

### Switching rate under non-activating conditions

In this work, the difference between the growth rates of the two morphotypes, $\varepsilon = \mu_B - \mu_s$, is small compared to the growth rates as determined by time-lapse microscopy: $\mu_B$=1.24 ± 0.02 h$^{-1}$ (N = 98), $\mu_S$=1.18 ± 0.02 h$^{-1}$ (N = 56), and $\varepsilon$ = 0.06 ± 0.03 h$^{-1}$ (means ± s.e.). Microscopic observations revealed that *b* is smaller than all other rates. In the limit of *b* = 0, *Equation 1* can be greatly simplified, and, when starting from the BIG morphotype (i.e., $S_0$ = 0), as done in the experiment of *Figure 2A–E*, we expect:

$$\frac{S(t)}{B(t)} = \frac{a}{\epsilon - a}\left[1 - e^{-(\epsilon - a)t}\right] \tag{2}$$

Starting from the BIG morphotype ($S_0$ = 0), we measured at t = 17 hr $\frac{S(t)}{B(t)}$=0.26 ± 0.04. Fitting *Equation 2* to the only free parameter, *a*, results in *a* = 0.021 (c.i. 0.017, 0.024) h$^{-1}$. Starting from the SMALL morphotype ($B_0$ = 0), we measured at t = 17 hr $\frac{S(t)}{B(t)}$=24 ± 18. Using the value of *a*

determined above, and fitting *Equation 1* to the only free parameter, $b$, results in $b$ = 0.0017 (c.i. 0.001, 0.007) $h^{-1}$.

## Switching rate under activating conditions

The switching rates under activating conditions (*Figure 3B*) were determined by monitoring the dynamics of formation of SMALL and BIG colonies from a culture containing only BIG morphotypes, as well as the growth of the whole culture (*Figure 3C*). Fitting the experimental data using a 2-parameter fit for *Equation 2* yields: $a$ = 0.24 (c.i. 0.10, 0.37) $h^{-1}$, $\Delta = a - \varepsilon = 0.21$ (c.i. 0.10, 0.31) $h^{-1}$.

Note that the key parameter that controls the time-scale of population dynamics is $\Delta = a - \varepsilon$, the difference between the loss of BIG due to switching and gain in BIG due to faster growth of BIG vs. SMALL bacteria. Whereas this parameter is positive under activating conditions, leading to a dominance of the SMALL morphotype at steady state, it has an opposite sign under non-activating conditions, predicting a stable co-existence of the two morphotypes even when $b$ = 0.

In summary, the mathematical model of switching enabled us to predict the population dynamics of the SMALL and BIG sub-populations in non-activating and in activating conditions. The main parameter that was found to change is $a$, the switching rate from BIG to SMALL, which increases more than 10-fold in activating conditions. Finally, the switching rate from SMALL to BIG, $b$, was found to be extremely low compared to all other time-scales, enabling a long-term memory of the SMALL morphotype over hundreds of generations.

## Acknowledgements

We thank Naomi Melamed-Book for assistance with confocal microscopy, William Breuer for assistance with FACS, Calin Guet, James Kaper, Michael Donnenberg, and Gad Frankel for plasmids, strains and antibodies, Erez Mills for construction of *perC::kan* EPEC. Benjamin Aroeti, Ady Vaknin, Sivan Pearl Mizrahi and Gad Frankel for comments on the manuscript. NQB is supported by the European Research Council (Consolidator Grant no. 681819), the Israel Science Foundation grant no. 492/15 and the Minerva Foundation. IR acknowledges funding by the Israel Science Foundation grant no 617/15. NQB holds the Joseph and Sadie Danciger Chair in Physics. NK is a recipient of a fellowship from the Carol and Leonard Berall Endowment. IR is an Etta Rosensohn Professor of Bacteriology.

## Additional information

### Funding

| Funder | Grant reference number | Author |
| --- | --- | --- |
| European Research Council | 681819 | Nathalie Q Balaban |
| Israel Science Foundation | 492/15 | Nathalie Q Balaban |
| Minerva Foundation | | Nathalie Q Balaban |
| Israel Science Foundation | 617/15 | Ilan Rosenshine |

The funders had no role in study design, data collection and interpretation, or the decision to submit the work for publication.

### Author contributions

IRon, Conceptualization, Resources, Data curation, Formal analysis, Validation, Investigation, Visualization, Methodology, Writing—original draft, Writing—review and editing; NK, Validation, Methodology, Acquisition of data; IRos, Conceptualization, Resources, Data curation, Supervision, Validation, Methodology, Writing—original draft, Writing—review and editing; NQB, Conceptualization, Resources, Data curation, Software, Formal analysis, Supervision, Funding acquisition, Validation, Visualization, Methodology, Writing—original draft, Writing—review and editing

### Author ORCIDs

Nathalie Q Balaban, http://orcid.org/0000-0001-8018-0766

## Additional files

### Supplementary files

• Supplementary file 1. Whole genome sequencing of the BIG and SMALL morphotypes. The relative copy number was obtained by coverage analysis of the whole genome sequencing data for BIG and SMALL bacteria (*Goldberg et al., 2014*). The WGS analysis reveals differences in EAF plasmid copy number only and the raw data is available as NCBI BioProject PRJNA255355 (Accessions: SRX757584 and SRX757585 for SMALL and BIG respectively).

• Supplementary file 2. List of strains and plasmids.

• Supplementary file 3. List of primers used in this study.

### Major datasets

The following previously published dataset was used:

| Author(s) | Year | Dataset title | Dataset URL | Database, license, and accessibility information |
|---|---|---|---|---|
| Goldberg A | 2014 | Escherichia coli Detection of Phase Variable Inversions | https://www.ncbi.nlm.nih.gov/bioproject/?term=PRJNA255355 | Publicly available at the NCBI BioProject (accession no: PRJNA255355) |

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
