## [Decision Letter]

Thank you for submitting your article "A Long-term Epigenetic Memory Switch Controls Bacterial Virulence Bimodality" for consideration by *eLife*. Your article has been reviewed by two peer reviewers, including Jose Luis Puente (Reviewer #3), and the evaluation has been overseen by a Reviewing Editor and Richard Losick as the Senior Editor.

The reviewers have discussed the reviews with one another and the Reviewing Editor has drafted this decision to help you prepare a revised submission.

Summary:

The reviewers agreed that this is a very nice paper that describes phenotypic variability in the EPEC strain E2348/69, demonstrating that the variability is likely associated with the plasmid-encoded PerA transcriptional regulator. The variable phenotype involves both growth rate as well as virulence gene expression. A particularly striking feature of the variability described here is the remarkable stability of the small morphotype subpopulation under non-activating conditions. The authors present a thorough analysis of the phenomenon and have taken the first steps towards establishing a mechanism through their identification of PerA as playing a central role. Although there was enthusiasm among the reviewers and reviewing editor, there were also some concerns. First, the paper makes little headway in establishing how PerA bimodality is achieved in a population. Understanding the origins of bimodality would elevate the paper's impact substantially, although it was agreed that a full understanding of this would be beyond the scope of a single paper. Second, and perhaps more importantly, there were concerns about whether the data unequivocally support the conclusion that PerA is the sole arbiter of the phenotypic variability. This issue, in particular, will need to be fully addressed in a revised manuscript – please see the revisions below for details.

Essential revisions:

The authors have not fully ruled out that PerB may also play a role in controlling bimodality as the *perA* deletion (and insertion of a kanR cassette) may have a polar effect on *perB*. Additionally, the *perA* mutant was complemented by putting back a plasmid with the entire perABC operon, and a *perB* deletion mutant was not tested. Also, the bimodal expression of the perABC operon is in agreement with colony-size bimodality, but the experiments performed don't rule out the possibility that something else on the plasmid or the chromosome contributes to phenotypic variability. Thus, a revised manuscript should:

1) Perform a complementation experiment as shown in the bottom middle panel of Figure 4 but using a plasmid that expresses only *perA*, instead of perABC. This will establish that the loss of bimodality in the *perA* deletion is due to the lack of PerA and not due to a change in PerB expression.

2) Also, perform a complementation experiment, as in Figure 4, in which the plasmid expressing *perA* alone is put into a strain that lacks the EAF plasmid, to determine whether PerA alone or through other PerA-regulated component of the EAF plasmid, is required for bimodality.

3) If PerA alone fails to fully complement in 1) then the two complementation experiments above should be done with PerB alone.

---

## [Author Response]

*Summary:*

*The reviewers agreed that this is a very nice paper that describes phenotypic variability in the EPEC strain E2348/69, demonstrating that the variability is likely associated with the plasmid-encoded PerA transcriptional regulator. The variable phenotype involves both growth rate as well as virulence gene expression. A particularly striking feature of the variability described here is the remarkable stability of the small morphotype subpopulation under non-activating conditions. The authors present a thorough analysis of the phenomenon and have taken the first steps towards establishing a mechanism through their identification of PerA as playing a central role. Although there was enthusiasm among the reviewers and reviewing editor, there were also some concerns. First, the paper makes little headway in establishing how PerA bimodality is achieved in a population. Understanding the origins of bimodality would elevate the paper's impact substantially, although it was agreed that a full understanding of this would be beyond the scope of a single paper. Second, and perhaps more importantly, there were concerns about whether the data unequivocally support the conclusion that PerA is the sole arbiter of the phenotypic variability. This issue, in particular, will need to be fully addressed in a revised manuscript – please see the revisions below for details.*

We completely agree that we had not dissected the respective roles of the *per* genes in our previous version, and wrongly had written *perA* where in fact we meant the whole *per* operon. We have now completed a set of experiments that follow the guidelines of the reviewers and show that *perA* expression alone does not result in bimodal growth, but bimodality is observed when both *perA* and *perB* are expressed. We have performed the experiments with similar results in two strains: an EPEC strain deleted for *perA* and complemented with either *perA, perB* or both *perA* and *perB*, and in *E. coli* K-12 MG1655 bearing the same set of plasmids. In both strains *perA*+*perB* resulted in bimodal growth whereas *perA* or *perB* alone did not.

We have now included these results in the fully revised version of the manuscript, as well as addressed all the additional comments of the reviewers, as detailed below.

*Essential revisions:*

*The authors have not fully ruled out that PerB may also play a role in controlling bimodality as the perA deletion (and insertion of a kanR cassette) may have a polar effect on perB. Additionally, the perA mutant was complemented by putting back a plasmid with the entire perABC operon, and a perB deletion mutant was not tested. Also, the bimodal expression of the perABC operon is in agreement with colony-size bimodality, but the experiments performed don't rule out the possibility that something else on the plasmid or the chromosome contributes to phenotypic variability. Thus, a revised manuscript should:*

*1) Perform a complementation experiment as shown in the bottom middle panel of Figure 4 but using a plasmid that expresses only perA, instead of perABC. This will establish that the loss of bimodality in the perA deletion is due to the lack of PerA and not due to a change in PerB expression.*

We thank the reviewer for this comment. To find out whether PerA is sufficient for the bimodality of growth that we observed, we transformed the Δ*perA* mutant with plasmid expressing PerA-GFP controlled by the native regulatory region, implying that PerA is required for expression. This design was essential to test whether PerA can drive bimodality through its autoregulation. However, complementation of Δ*perA* with PerA-GFP expressing plasmids resulted in unimodal colony morphotypes (Figure 5), that were found to have a uniformly high expression of pPerA-GFP (Figure 5). These results show that PerA is not sufficient for growth bimodality (subsection “PerA and PerB are sufficient for the bimodal phenotype” and Figure 5).

*2) Also, perform a complementation experiment, as in Figure 4, in which the plasmid expressing perA alone is put into a strain that lacks the EAF plasmid, to determine whether PerA alone or through other PerA-regulated component of the EAF plasmid, is required for bimodality.*

To answer this question we used K-12 *E. coli* MG1655 (which lacks EAF plasmid). Similarly, to the results obtained with the Δ*perA* EPEC mutant, transformation with pPerA-GFP results in unimodal growth (Figure 5—figure supplement 2). However, transformation with pPerAB- GFP did result in bimodal growth (see below), showing that the EAF plasmid is not required for the *per* mediated bimodality of growth. We note that the bimodal phenotype is milder in the MG1655 strain. Results of this experiment are described now in the subsection “PerA and PerB are sufficient for the bimodal phenotype” and presented in Figure 5—figure supplement 2.

*3) If PerA alone fails to fully complement in 1) then the two complementation experiments above should be done with PerB alone.*

As predicted by this reviewer, we found that PerA alone fails to restore bimodality. Therefore, we repeated the two complementation experiments with the pPerB-GFP plasmid. In addition, we also measured the complementation with pPerAB-GFP, both constructs under the control of the native *per* regulatory region.

The complementation assays of the Δ*perA* mutant with plasmids expressing pPerB- GFP and pPerAB-GFP are now shown in Figure 5. Whereas complementation of PerB-GFP expressing plasmids resulted in unimodal colony morphotypes, the PerAB- GFP plasmid restored bimodality, indicating that PerA and PerB are sufficient for the bimodality. Microscopic observations show that GFP expression is bimodal in pPerAB-GFP plasmids, while PerB and GFP cannot be expressed from pPerB-GFP in the absence of PerA and, accordingly, no GFP was observed in Δ*perA* mutant containing this plasmid (Figure 5). Expression of GFP from this plasmid was restored in the *wt* EPEC (Figure 5—figure supplement 1). These results show that *perA* and *perB* expression are required for the co-existence of the BIG and SMALL morphotypes. We have now added description and results for this experiment in the subsection “PerA and PerB are sufficient for the bimodal phenotype”.

The complementation assays of the MG1655 strain with plasmids expressing pPerB- GFP and pPerAB-GFP are now shown in Figure 5—figure supplement 2. The results are very similar to the results obtained with the Δ*perA* mutant, i.e. PerA and PerB together restore the bimodality of growth. These results suggest that EAF plasmid factors are not essential for the growth bimodality. We note that the growth bimodality is milder in the MG1655 strain. Description and results for this experiment are in the aforementioned subsection.